# Highly sensitive and broadband meta-mechanoreceptor via mechanical frequency-division multiplexing

Chong Li[1], Xinxin Liao[1], Zhi-Ke Peng[1,2], Guang Meng[1] & Qingbo He ®[1]✉

Bio-mechanoreceptors capable of micro-motion sensing have inspired mechanics-guided designs of micro-motion sensors in various fields. However, it remains a major challenge for mechanics-guided designs to simultaneously achieve high sensitivity and broadband sensing due to the nature of resonance effect. By mimicking rat vibrissae, here we report a metamaterial mechanoreceptor (MMR) comprised of piezoelectric resonators with distributed zero effective masses featuring a broad range of local resonances, leading to near-infinite sensitivity for micro-motion sensing within a broad bandwidth. We developed a mechanical frequency-division multiplexing mechanism for MMR, in which the measured micro-motion signal is mechanically modulated in non-overlapping frequency bands and reconstructed by a computational multi-channel demodulation approach. The maximum sensitivity of MMR is improved by two orders of magnitude compared to conventional mechanics-guided mechanoreceptors, and its bandwidth with high sensitivity is extendable towards both low-frequency and high-frequency ranges in 0–12 kHz through tuning the local resonance of each individual sensing cell. The MMR is a promising candidate for highly sensitive and broadband micro-motion sensing that was previously inaccessible for mechanics-guided mechanoreceptors, opening pathways towards spatio-temporal sensing, remote-vibration monitoring and smart-driving assistance.

Nature has found a way to achieve remarkable micro-motion sensing by evolving biological mechanoreceptors[1]. Biological mechanoreceptors, such as mouse vibrissae[2], fish lateral lines[3], spider slits[4], spider silk[5], scorpion crack[6], and human skin[7,8], are capable of detecting ubiquitous micro-motions with high sensitivity from a few hertz to thousands of hertz. Inspired by these naturally existing mechanoreceptors, artificial mechanoreceptors that convert mechanical micro-motions into electrical signals, have been emerging and widely used in bioengineering[9,10], medical diagnosis[11,12], military defense[13,14], ocean exploration[15,16], Internet of Things[17,18], robotics[19,20], and electronics[8]. Bioinspired designs of artificial mechanoreceptor provide an efficient way to improve the performance of engineered

sensors, which can accelerate the development of next-generation intelligent sensing systems[21,22], aiming at high precision, low cost, and low complexity.

For the design of artificial mechanoreceptors capable of micro-motion sensing, the sensitivity and bandwidth are critical performance metrics and have always been the focus of sensor-related research[5,23]. Artificial mechanoreceptors are classified into two main types based on design strategies to improve the sensitivity and bandwidth. One type is the electro-guided design relying on highly sensitive electrical elements[24] which suffer from high-cost and complex nanofabrication processes, such as nanofibers[25] and nanowires[26]. The other type is the mechanics-guided design, which optimizes the electromechanical

[1]State Key Laboratory of Mechanical System and Vibration, Shanghai Jiao Tong University, Shanghai 200240, P. R. China. [2]School of Mechanical Engineering, Ningxia University, Yinchuan 750021, P. R. China. ✉e-mail: qbhe@sjtu.edu.cn

coupling factor with low-cost and simple mechanical structures[27] with rationally designed resonances, such as chaotic oscillator[28] and buckling architecture[29]. However, there's a trade-off between the sensitivity and bandwidth of mechanics-guided designs due to the nature of resonance effect[30–32].

Metamaterials have proven to be extraordinary artificial structures with advanced mechanics-guided designs, and exhibit exotic physical properties beyond naturally occurring structures[33,34]. Due to their pre-programmable effective properties and functionalities[35], they have been widely applied to metamaterial-enhanced sensors for acoustic waves[36], optical signals[37], and thermal transmissions[38]. However, the research on metamaterial-enhanced micro-motion sensing remains elusive[39], which is limited to strain measurements within a narrow bandwidth[40–42]. To date, it is still challenging to simultaneously achieve highly sensitive and broadband micro-motion sensing with mechanics-guided metamaterial designs, which is in high demand for development of high-performance micro-motion sensors.

Inspired by the remarkable micro-motion sensing of rat vibrissae, we developed a metamaterial mechanoreceptor (MMR) to address this challenge. The designed MMR consists of piezoelectric resonators with distributed zero effective masses in different frequencies, resulting in highly sensitive and broadband micro-motion sensing. Through analytical, numerical, and experimental verifications, we demonstrate that these piezoelectric resonators with zero effective masses exhibit frequency-dependent effective piezoelectric coefficients, which are not achievable in intrinsic piezoelectric materials and lead to near-infinite sensitivity within a broad bandwidth. We tessellated these piezoelectric resonators as unit cells into a metamaterial architecture that performs as "meta-mechanoreceptor" for micro-motion sensing. We developed a mechanical frequency-division multiplexing system to mechanically modulate micro-motion signals in non-overlapping frequency bands, which combined a computational multi-channel demodulation approach to achieve the signal reconstruction. Compared with the conventional piezoelectric mechanoreceptor with a sensitivity of 0.812 mv m$^{-1}$s$^2$, MMR has a maximum sensitivity that is 2 orders of magnitude higher, reaching up to 36.540 mv m$^{-1}$s$^2$, and the lowest detection limit is estimated to reach 10$^{-6}$ g in the order of magnitude. Through integrating sensing units with prescribed piezoelectric coefficients, the bandwidth of MMR can be extended towards both low frequencies and high frequencies in 0–12 kHz, leading to broad applications in spatio-temporal sensing, remote-vibration monitoring, and smart-driving assistance. The designed MMR is a promising candidate for highly sensitive and broadband micro-motion sensing that has not been attainable in conventional mechanics-guided micro-motion sensors, breaking the trade-off between sensitivity and bandwidth in the state-of-art mechanoreceptors with single-piece piezoelectric elements.

## Results

### Design of vibrissae-inspired meta-mechanoreceptor

Rats' vibrissae system is capable of detecting micro-motions with high sensitivity and broad bandwidth, making it possible to sense micro-motions in a noisy environment[43] (Fig. 1a, b). Each vibrissa is a highly sensitive mechanoreceptor and its length ($L$) results in a specific working frequency range[44] (Fig. 1c). An array of vibrissae featuring different lengths leads to the broadband micro-motion sensing[45]. When functioning, the vibrissae mechanically enhances the micro-motions with the resonance effect and then convert the micro-motions into neural potentials to transmit via the afferent nerves (Supplementary Video 1).

We designed the bioinspired piezoelectric resonator as a unit cell consisting of a spiral base, a piezoelectric stack, and a copper pillar (Fig. 1d). The micro-motion excitations trigger the locally resonant state[46] of the unit cell to enhance the piezoelectric charge output in a frequency range around its resonance frequency, which contributes to

high sensitivity for micro-motion sensing. We tailor the angle $\theta_n$ of the spiral base to tune the local resonance frequency of each individual sensing cell (Fig. 1e). We tessellate the unit cells with different $\theta_n$ into a plane metamaterial architecture for micro-motion sensing, which is denoted as MMR (Fig. 1f). The MMR features a broad range of local resonances under micro-motion excitations, and exhibits different charge output at different frequencies, leading to frequency-dependent effective piezoelectric coefficients. Similar to the neural potential processing in rats' vibrissae system, we developed an artificial deep neural network to process the signal output of the MMR.

The ubiquitous micro-motions in nature contain rich physical information (e.g., space-time patterns, rhythmic musical notes, or vehicle health status), which are quite weak and difficult to be measured in extreme environment (Fig. 1g). The working frequency range of conventional piezoelectric mechanoreceptors is in the non-resonant range far from the resonance frequency. The broadband micro-motion excitations are not enhanced and thus difficult to be detected (Fig. 1h). On the contrary, the unit cells with localized resonance frequencies in the MMR contributes to frequency-dependent effective piezoelectric coefficients with sensitivity-enhanced micro-motion sensing, and the operating band can be extended towards both low frequencies and high frequencies to fully cover the non-resonant frequency band in conventional piezoelectric mechanoreceptors.

### Frequency-dependent effective piezoelectric coefficient

The MMR combines unit cells with mechanics-guided piezoelectric resonator design to bypass the limitations of intrinsic piezoelectric materials and achieve high sensitivity. In conventional piezoelectric materials, the electromechanical conversion is described by existing piezoelectric tensors defined under quasi-static conditions. As an example, $d_{33}$ describes the correlation between the stress and the electric displacement in the intrinsic piezoelectric material along the z-direction in Cartesian coordinates[47]. However, the $d_{33}$ defined under quasi-static constraints is not sufficient to describe the piezoelectric property of the unit cell with zero effective mass under dynamic frequency-varying conditions. We define the frequency-dependent effective piezoelectric coefficient $d_{33\text{eff}}(\omega)$ based upon elastodynamics theory[48] to describe the electromechanical conversion of the unit cell in different frequencies as follows (Supplementary Note 1 and Supplementary Fig. 1):

$$d_{33\text{eff}}(\omega) = A(\omega)d_{33}\frac{1}{m_{\text{eff}}(\omega)} \tag{1}$$

where $A(\omega)$ represents a frequency-dependent variable related to the dynamic parameter of the unit cell, and $\omega$ denotes the varying frequency. With the analytical method (Fig. 2a and Supplementary Note 1), we derived the effective mass $m_{\text{eff}}(\omega)$ of the unit cell, and found that $m_{\text{eff}}(\omega)$ was equal to zero at a specific frequency that was denoted by zero-mass frequency (Fig. 2b). Remarkably, we found that the $d_{33\text{eff}}(\omega)$ was theoretically a near-infinite value at the zero-mass frequency, which has not been demonstrated in existing piezoelectric tensors.

Finite element simulation of the unit cell was performed to verify the theoretical predictions (see "Methods" section for details). As indicated by the dispersion curves (Fig. 2c), the vibration mode at the zero-mass frequency was corresponding to the locally resonant mode (denoted by M1). When the unit cell was excited by external micro-motions (finite $F$) at the zero-mass frequency, the dynamic response was enhanced by the zero effective mass ($m_{\text{eff}}(\omega) \longrightarrow 0$) with a near-infinite acceleration amplitude ($F/m_{\text{eff}}(\omega) \longrightarrow \infty$). We applied micro-motion excitation to the unit cell in different directions, and confirmed that the unit cell was mono responded in $d_{33}$ direction (Supplementary Note 4 and Supplementary Figs. 3 and 4). We calculated the $m_{\text{eff}}(\omega)$ and $d_{33\text{eff}}(\omega)$ of the unit cell with the

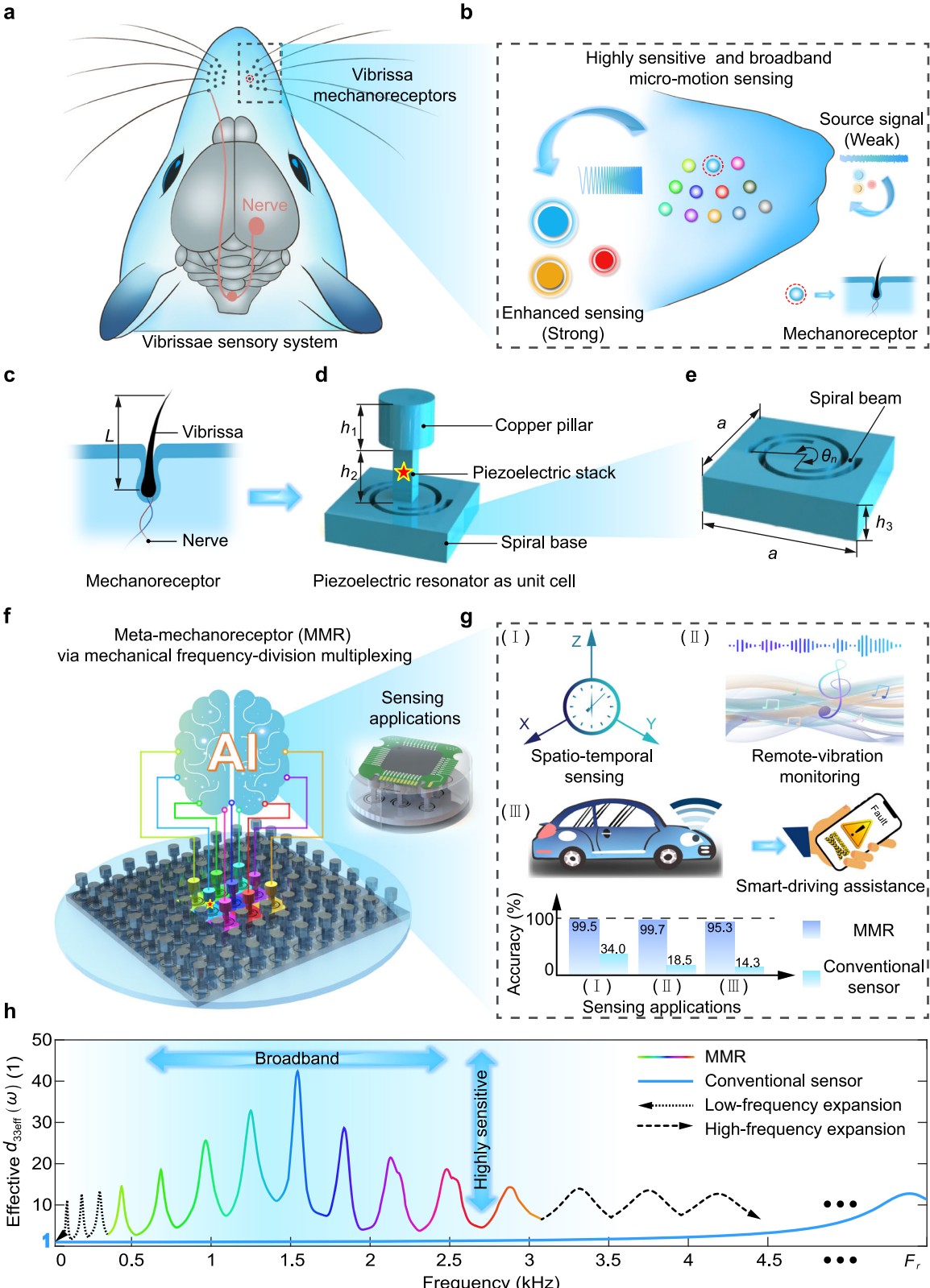

**Fig. 1 | Vibrissae-inspired meta-mechanoreceptor (MMR). a** Rats' vibrissae system is capable of detecting micro-motions in a noisy environment. **b** Each vibrissa with the length $L$ is a highly sensitive mechanoreceptor in a specific frequency range, and an array of vibrissae featuring different lengths leads to the broadband micro-motion sensing. **c** The micro-motions are enhanced by the resonance of the vibrissa and then converted into neural potentials via the afferent nerves. **d** Vibrissae-inspired piezoelectric resonator as a unit cell with (**e**) a spiral angle of $\theta_n$.

**f** The MMR that is tessellated by unit cells with different $\theta_n$ behaves as a mechanical frequency-division multiplexing system. **g** The micro-motion recognition with MMR is superior to that with the conventional piezoelectric mechanoreceptor (combination of a piezoelectric stack and a copper pillar only). **h** The MMR exhibits frequency-dependent effective piezoelectric coefficients (normalized) with sensitivity-enhanced micro-motion sensing, and the working band can be extended in both low frequencies and high frequencies.

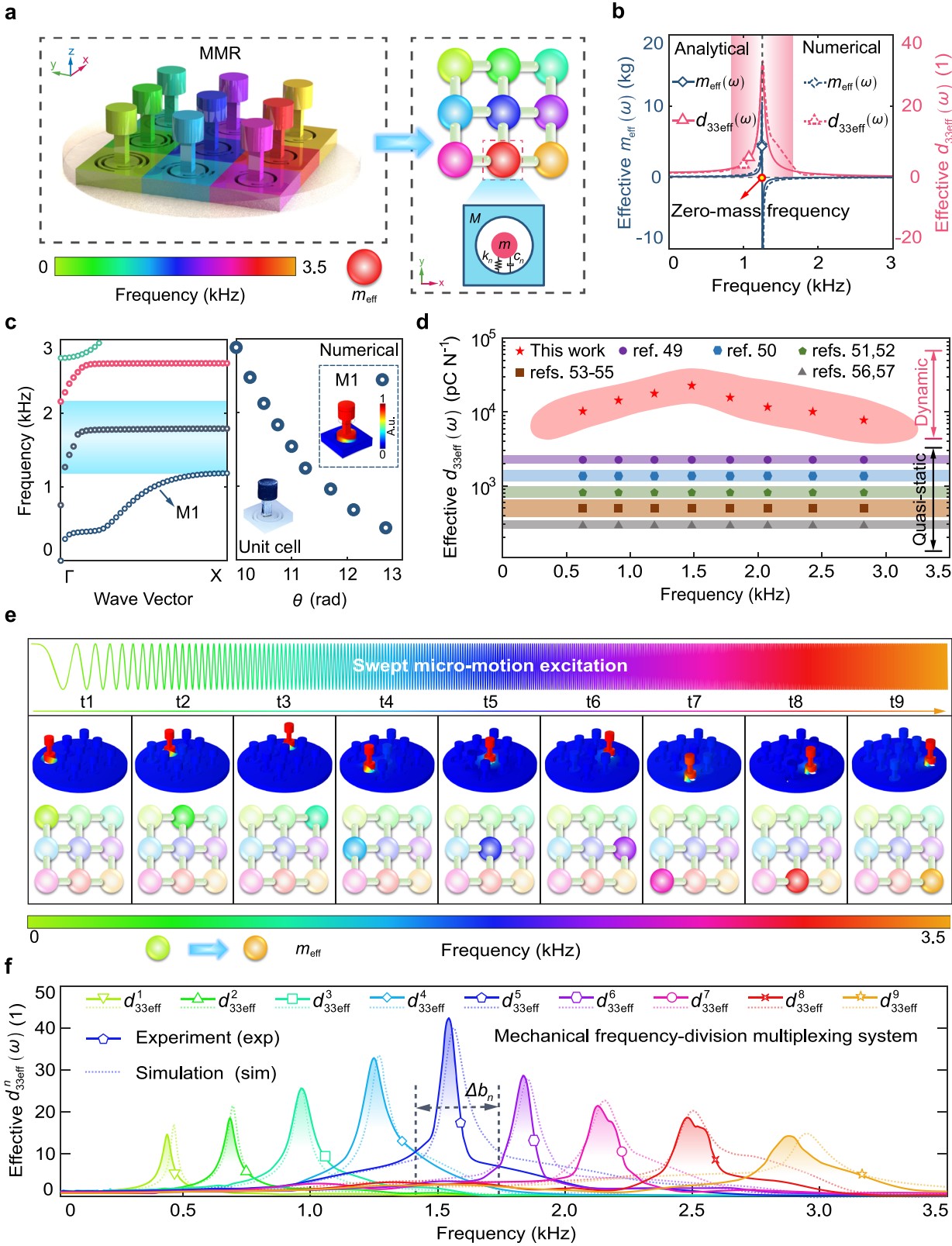

numerical method, which showed excellent agreement with the analytical results (see the red area in Fig. 2b). Due to the inevitable damping in experiments, the maximum $d_{33eff}(\omega)$ reached an appreciable value of 24,930 C N$^{-1}$ instead of a theoretical near-infinite value (Fig. 2d), which was much higher than that of the state-of-art piezoelectric materials[49–57]. Therefore, configuring zero effective mass design in the unit cell results in a near-infinite effective

piezoelectric coefficient, allowing access to micro-motion sensing with a high sensitivity around the zero-mass frequency.

## Mechanical frequency-division multiplexing system

We developed a mechanical frequency-division multiplexing system as the physical mechanism of MMR (Supplementary Video 2). To explain this mechanism, we established a global dynamical model for MMR,

**Fig. 2 | Mechanical frequency-division multiplexing system. a** Schematic diagram of the designed MMR, in which the unit cells are simplified to local resonators, and the colors indicate the zero-mass frequencies. **b** Frequency-dependent effective mass and effective piezoelectric coefficient (normalized) of the unit cell ($\theta_n = 11.25$) with the analytical derivation and numerical simulation. The effective piezoelectric coefficient of the unit cell reaches near-infinite at the zero-mass frequency, leading to the ultrahigh sensitivity in a specific frequency range. **c** Dispersion curves of the unit cell with $\theta_n = 11.25$. The zero-mass frequency value

corresponding to the M1 mode decreases with the increase of $\theta_n$. **d** Comparison of effective piezoelectric coefficient between the unit cell under dynamic frequency-varying conditions in this study and typical piezoelectric materials under quasi-static conditions. **e** The locally resonant responses of MMR under swept micro-motion excitation. **f** Enhanced micro-motion responses in a series of non-overlapping frequency bands, which makes MMR a mechanical frequency-division multiplexing system for highly sensitive and broadband micro-motion sensing.

and derived the effective piezoelectric coefficient $d_{33\text{eff}}^n(\omega)$ of the $n$th unit cell in this model as follows (Supplementary Note 5):

$$d_{33\text{eff}}^n(\omega) = A(\omega)d_{33}\frac{1}{M_{\text{eff}}(\omega)} \quad (2)$$

where $M_{\text{eff}}(\omega)$ is the frequency-dependent effective mass of MMR and generates distributed zero values in different frequencies (Supplementary Fig. 5). The MMR exhibits near-infinite $d_{33\text{eff}}^n(\omega)$ at these distributed zero-mass frequencies. We applied swept micro-motion excitation to MMR, and observed that MMR featured a broad range of local resonances around the zero-mass frequencies (Fig. 2e). This feature led to enhanced micro-motion responses in non-overlapping frequency bands denoted by $\Delta b_n$ (Fig. 2f), which enabled the MMR to behave as a mechanical frequency-division multiplexing system, achieving the high sensitivity within a broad bandwidth. By individually tuning the $\theta_n$ of the unit cell, the $\Delta b_n$ allowed for a tailored design of zero-mass frequency value among 0–12 kHz (Supplementary Fig. 6).

In this mechanical frequency-division multiplexing system, MMR provided a mechanical transmission platform for micro-motion excitations, and the total transmission bandwidth was divided into a series of non-overlapping frequency bands, each of which was exploited to carry modulated and enhanced micro-motion responses around the specific zero-mass frequency. Here, the response signal initiated by the micro-motion excitation was directly modulated via the zero-mass metamaterials, which eliminated the need for filters, mixers, and amplifiers commonly used in electrical frequency-division multiplexing systems[58]. Therefore, the MMR offered a low-cost and low-complexity design for implementation of high-performance micro-motion sensors, which facilitated the highly sensitive and broadband property.

## Computational multi-channel demodulation of the measured signal

With the mechanical frequency-division multiplexing mechanism in MMR, the micro-motion-induced voltage output $V_n(\omega)$ within a broad bandwidth (denoted by $B = \sum \Delta b_n$) was measured simultaneously through $N$ unit cells (Fig. 3a). Each unit cell acted as a narrowband measurement channel, which modulated the micro-motion response nonlinearly in the specific band $\Delta b_n$, and gave rise to different voltage outputs in different frequency range. Due to the nonlinear modulation, the main technical challenge associated with the implementation of MMR lay in the ability to acquire the original micro-motion signal from the multi-channel measurements. We mimicked rats' neural system and developed a computational multi-channel demodulation approach for signal reconstruction (Supplementary Note 6 and Supplementary Video 3). When functioning (Fig. 3b), the measured voltage $V_n(\omega)$ in each channel was processed in combination with the pre-calibrated frequency responses $H_n(\omega)$, which was to reconstruct the original micro-motion signal $S(\omega)$ through solving the inverse problem, $S(\omega)H_n(\omega) = V_n(\omega)$. This approach critically determined the measurement accuracy, especially for micro-motion sensing with a low signal-to-noise ratio (SNR) when a background noise surpassed the original signal heavily.

To evaluate the sensing performance of MMR, we constructed an experimental sensing system consisting of 9 measurement channels in

the as-fabricated $3 \times 3 \times 1\,\text{cm}^3$ MMR (see "Methods" section). We carried out experiments with two different types of micro-motion signals in the selected band of 0–3.5 kHz. First, we constructed harmonic signals with nine frequency components of equal amplitude as the multi-frequency signal. From the measured results (Fig. 3c), the maximum sensitivity of MMR was improved by two orders of magnitude from 0.812 mv m$^{-1}$s$^2$ to 36.540 mv m$^{-1}$s$^2$ as compared to the conventional piezoelectric mechanoreceptor. The measurement accuracy can be further improved through multiple repeated measures (Supplementary Note 7 and Supplementary Fig. 9). The lowest detection limit of MMR was demonstrated to reach $10^{-6}$ g in the order of magnitude (Supplementary Note 8 and Supplementary Fig. 10). Second, we conducted comparative experiments with a chirp signal as the broadband signal at SNR = 0 dB (Fig. 3d), from which the measured result with MMR presented clearer time-varying frequency components than that with the conventional piezoelectric mechanoreceptor. Moreover, when the SNR reaches −20 dB (Fig. 3e), the MMR can still measure the majority of the signal components, while the conventional piezoelectric mechanoreceptor cannot measure clear signal components since the chirp signal with SNR of −20 dB cannot reach the detection limit. We further demonstrated that MMR shows strong stability and reliability under long-time measurement of 1000 s and under the measurement of different SNRs from −30 dB to 10 dB (Supplementary Note 9, Supplementary Figs. 11 and 12). Compared to other methods (such as non-resonant[49,50], resonant[59,60], and biomimetic[11,61] designs), MMR addressed the trade-off between sensitivity and bandwidth, which was previously unachievable with the conventional piezoelectric mechanoreceptor (Supplementary Note 10 and Supplementary Tables 3 and 4), opening opportunities for perception, monitoring, and identification of environmental micro-motions.

## Spatio-temporal sensing, remote-vibration monitoring, and smart-driving assistance via MMR

We constructed an intelligent sensing system for broad applications by combining MMR with a deep learning technique (Fig. 4a). The micro-motion signal was measured by MMR and then was transported to a deep neural network to complete the signal recognition (Supplementary Note 11). As a proof-of-concept demonstration of spatio-temporal sensing (Supplementary Video 4), we constructed harmonic signals with multiple frequencies, leading to resonant responses of unit cells sequentially at different locations. A spatio-temporal pattern was observed by tracking the response locations of unit cells. We constructed four patterns "SJTU" based on four signals (Supplementary Fig. 14 and Supplementary Table 5). When the SNR was −20 dB, the recognition accuracies of patterns with and without MMR were about 99.5% and 34.0%, respectively (Fig. 4b). The recognition results confirmed that the MMR played a role in the in-situ enhancement of signal features (Supplementary Note 12 and Supplementary Fig. 15), which held great promise for building intelligent sensing systems to obtain high recognition accuracy, showing the application of MMR for identification and communication of micro-motion information in extreme environments.

We further demonstrated a remote-vibration monitoring approach with this intelligent sensing system. We showed a remote-vibration scene of a piano playing (Supplementary Video 5), where the waves carrying the message of rhythmic music spread throughout the

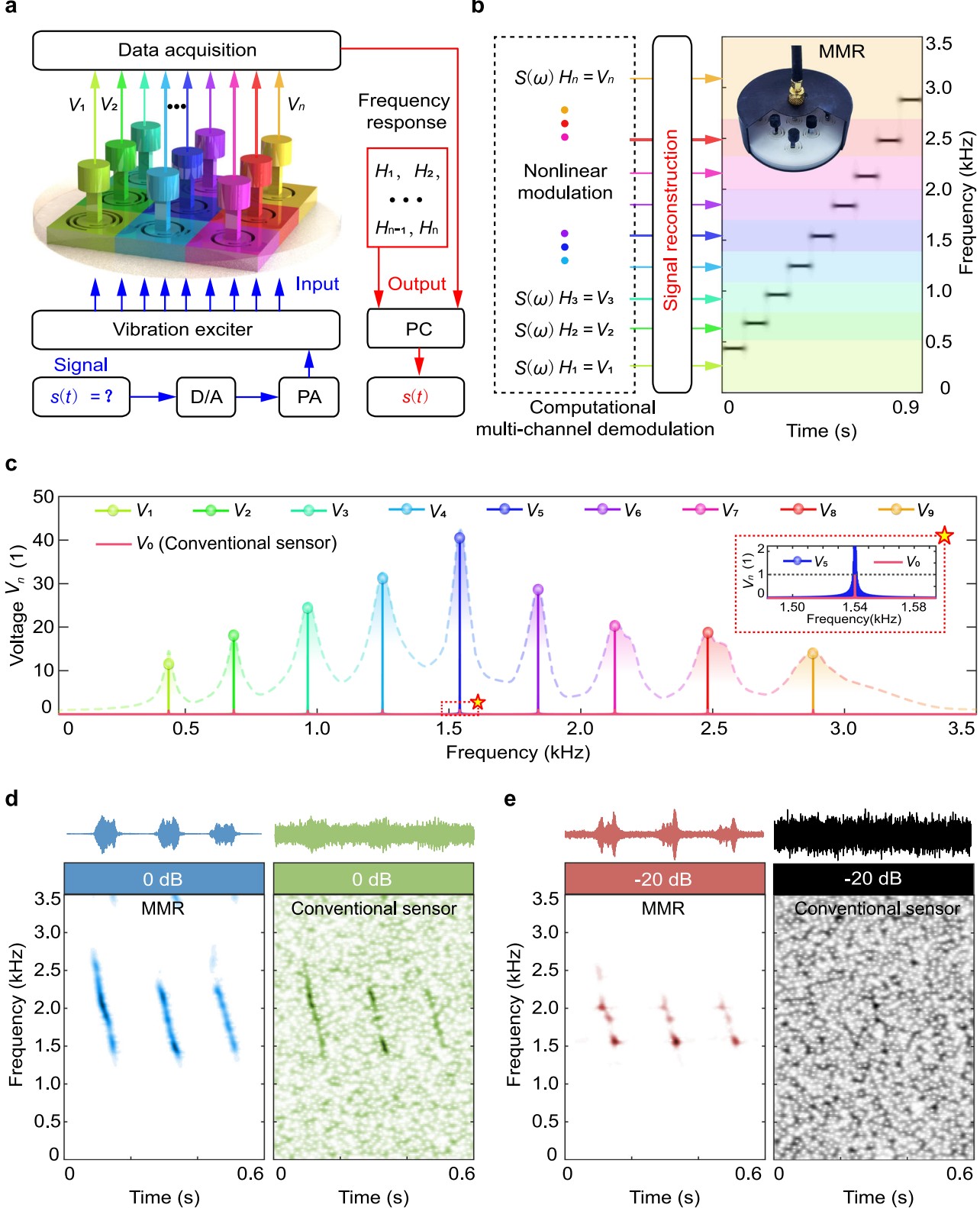

**Fig. 3 | Computational multi-channel demodulation of the measured signal.**
**a** The schematic of micro-motion sensing system with MMR. Each unit cell is a narrowband sensing channel with enhanced nonlinear modulation of the measured signal in a specific frequency range. **b** Schematic of the computational multi-channel demodulation through solving the inverse problem. The ladder-type signals in time-frequency representation are reconstructed successfully.
**c** Comparison of the measured multi-frequency signal (normalized) between the MMR and the conventional piezoelectric mechanoreceptor. **d** At the signal-to-noise rate (SNR) of 0 dB, the measured broadband signal with MMR is clearer than that with the conventional piezoelectric mechanoreceptor. **e** At the SNR of −20 dB, the conventional piezoelectric mechanoreceptor fails to work because the chirp signal has strong background noise and cannot reach the detection limit, while the MMR can still measure the majority of the signal components with the highly sensitive and broadband property.

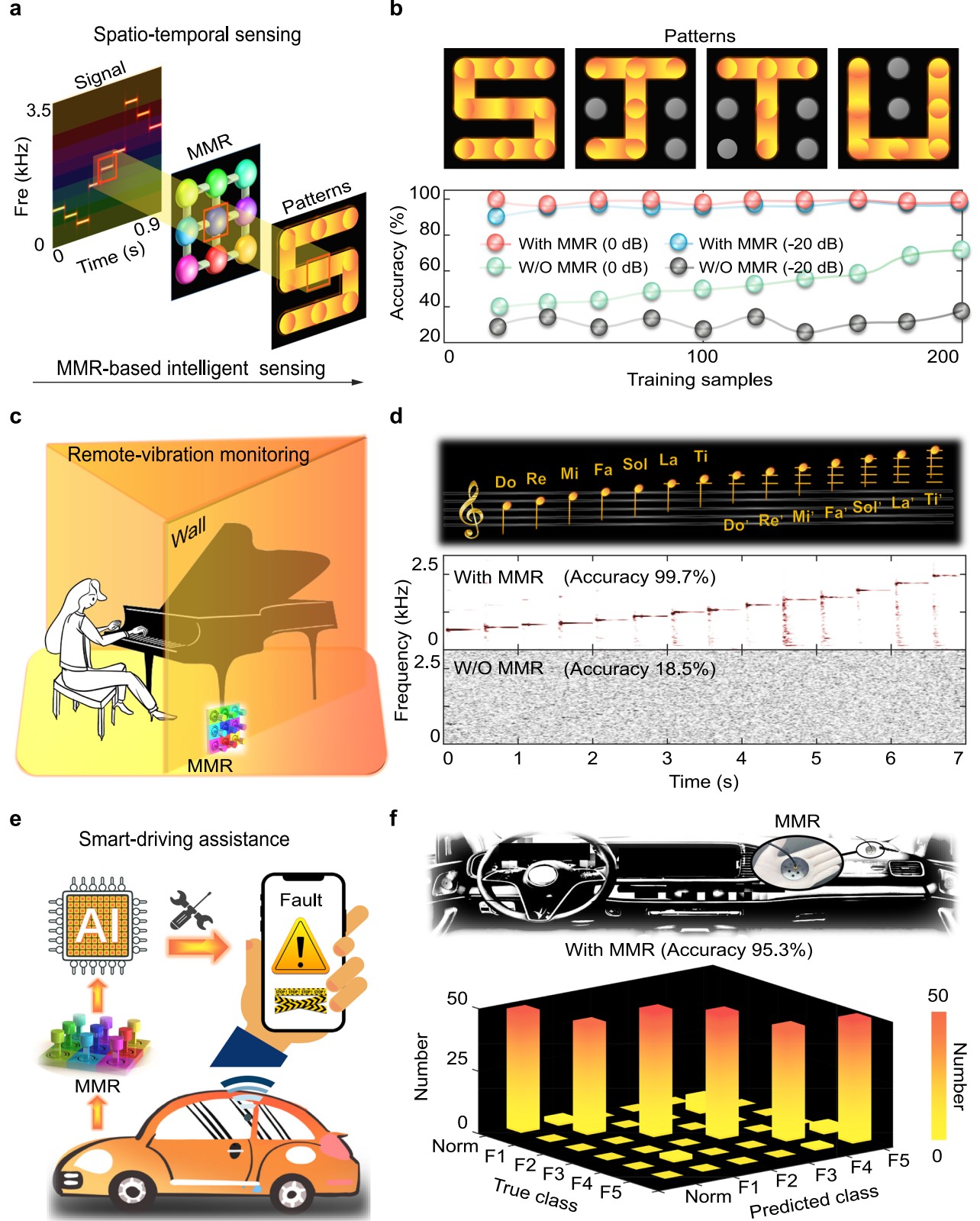

**Fig. 4 | Demonstration of spatio-temporal sensing, remote-vibration monitoring, and smart-driving assistance. a** Schematic of spatio-temporal sensing. The harmonic signals generate resonant responses of unit cells sequentially at different locations, forming a spatio-temporal pattern with in-situ enhanced sensing, which is further processed by a deep neural network to realize the pattern recognition. **b** The constructed patterns "SJTU" and comparison of the recognition accuracy with and without MMR. **c** Schematic of remote-vibration monitoring.

The piano playing is recognized by monitoring the weak vibrations on the exterior walls. **d** The recognition accuracies of musical scales with and without MMR are 99.7% and 18.5%, respectively. **e** Schematic of smart-driving assistance. **f** The average recognition accuracy of six engine signals (including one normal and five faults) with MMR is 95.3%, which is more than 5 times higher than the accuracy of 14.3% without MMR (Supplementary Note 15).

room and caused remote vibrations on the exterior wall (Fig. 4c). We identified the piano playing inside the room by monitoring the vibrations of the exterior wall. The working mechanism was that the vibration information of the wall was first extracted through the MMR, and then was processed in the deep neural network for identification of piano playing. We carried out identification experiments of 14 musical scales, including 7 Do-Ti scales and 7 higher Do'-Ti' scales. The real-time spectrogram of musical scales could be successfully detected by MMR, but was inaccessible without MMR due to the micro-vibrations corrupted by strong noise (Fig. 4d). We demonstrated that MMR recognized the musical scales with a high accuracy of 99.7%. In contrast, the recognition accuracy without MMR was 18.5%. Furthermore, we illustrated that MMR could record complex music information in real time by monitoring the remote vibrations, which was difficult to be achieved in the case without MMR (Supplementary Note 14 and Supplementary Figs. 18 and 19). The results indicated that MMR exhibited excellent detection abilities for remote-vibration monitoring, which could have potential for remote information acquisition and decryption in the field of security and confidentiality.

Moreover, we demonstrated a smart-driving assistance scene by equipping the MMR into an in-vehicle system (Supplementary Video 6), which completed signal collection, information processing and decision execution to ensure driving safety (Fig. 4e). We identified the health status of engine by monitoring the micro-motions on the cab platform far away from the engine. Early abnormal vibration signal from the engine is attenuated during transmission and difficult to be detected, resulting in a great safety hazard. Here, the early fault signal can be extracted with MMR in time, and is processed by the deep neural network to complete the identification of early faults, achieving autonomous monitoring and diagnosis. The vehicle executes the decisions based on the identification results, such as fault warning and safety inspection. We conducted experiments with six engine vibration signals, including one normal signal and five typical faulty signals from specific situations (Fig. 4f). The working conditions of the signals in the training set and the testing set are consistent, and the quality of signal acquisition is the core which directly determines the recognition accuracy. The recognition accuracy with MMR was 95.3%, which was more than 5 times higher than the accuracy of 14.3% without MMR (Supplementary Note 15 and Supplementary Figs. 20 and 21). Furthermore, we illustrated that MMR enabled structural health monitoring in the helicopter transmission system by detecting early rotor failures, which was unachievable in the case without MMR (Supplementary Note 16 and Supplementary Figs. 22–24). Therefore, the demonstrated results revealed that MMR was a promising candidate for highly sensitive and broadband sensing in the Internet of Vehicles system or other industrial scenarios.

## Discussion

We have reported a bioinspired MMR for highly sensitive and broadband micro-motion sensing via mechanical frequency-division multiplexing. MMR addresses the trade-off between sensitivity and bandwidth in conventional mechanics-guided micro-motion sensors. This is achieved by tessellating piezoelectric resonators with distributed zero effective masses into a metamaterial architecture that performs as MMR for micro-motion sensing. MMR exhibits frequency-dependent effective piezoelectric coefficients that break the limits of intrinsic piezoelectric materials in which the electromechanical conversion is described by existing piezoelectric tensors under quasi-static stations. As a result, the maximum sensitivity of MMR was improved by two orders of magnitude compared with the conventional piezoelectric mechanoreceptor, and its bandwidth with high sensitivity was extendable towards both low-frequency and high-frequency ranges in 0–12 kHz. As an application, we illustrate that MMR plays a role in the in-situ signal enhancement for building intelligent sensing systems, with capabilities for spatio-temporal sensing,

remote-vibration monitoring and smart-driving assistance. The rationally designed MMR can also be integrated into smart sensing platforms for highly sensitive and broadband micro-motion sensing in various fields, such as robotics and Internet of Vehicles. Overall, our work takes a significant step towards highly sensitive and broadband micro-motion sensing in mechanics-guided sensors, providing a perspective on simpler mechanics-guided designs of high-performance sensors for various other physical information.

Experimental results demonstrate a proof-of-concept application of MMR for micro-motion sensing. Nevertheless, our current study also has certain limitations. The application scene of MMR is suitable for single-point and single-axis measurement of micro-motions. The detection direction of MMR is limited to the $d_{33}$ direction. The MMR can also be used for detection of complex micro-motion that has a component projected in the $d_{33}$ direction. Further research on MMR can expand the spatial distribution of unit cells to three-dimensional configurations, which can be optimized for improved sensing performance in multiple directions[62]. The Lego-like heterogeneously constructed metamaterials provide a high degree of freedom for tailoring the sensitivity and bandwidth of MMR, which can be combined with stimuli-responsive metamaterials for adaptive sensing in different frequency ranges[63,64]. By using advanced micro and flexible manufacturing technology[65,66], the width and height of MMR can be made in subwavelength, which is hoped to be integrated into various flexible micro-electro-mechanical systems. By matching the integrated central processing unit (CPU), MMR can be packaged as an all-in-one system with sensing, storage, and computing functions (Supplementary Note 19 and Supplementary Fig. 27), which can enable the entire sensing system to be simpler and more portable[67–69]. MMR can also provide high versatility for complex micro-motion scenes with higher accuracy by incorporating advanced artificial intelligence (AI) techniques[70]. While our demonstrated results are within the mechanical sensing area, the proposed strategy could be extended to broad sensing fields with higher frequencies such as ultrasonic wave, millimeter wave, and terahertz bands, which may have a broad impact in designing various sensors[71].

## Methods
### Fabrication of MMR
The preparation of MMR began with the fabrication of unit cells. We determined the structural parameters through systematic parameter design combined with the existing fabrication technology. We first fabricated the spiral base of the unit cell with photosensitive resin. In order to ensure the fabrication accuracy and strength of the spiral beam while keeping the size as small as possible, we designed the parameters as $h_3 = 2.5$ mm and $a = 10$ mm. The maximum range of $\theta_n$ was 8.0–17.5 in a square base of 10 mm by 10 mm. Then we assembled the piezoelectric stack and the copper pillar to the spiral base as shown in Fig. 1d, which was to construct a piezoelectric resonator as a unit cell. For copper pillars and piezoelectric stacks, we designed the parameters as follows in combination with current commercial machining and sintering processes: $h_1 = 3.64$ mm and $h_2 = 5$ mm. The section of the piezoelectric stack was a square with a side length of 1.65 mm, and the section of the copper column was a circle with a radius of 2.2 mm. We fabricated unit cells with different $\theta_n$ and combined them into a scalable plane array as the MMR. In the experiments, the constructed MMR consisting of 3 × 3 unit cells was shown in Supplementary Fig. 7, and the parametric details were presented in Supplementary Table 1. The details of structural parametric design in MMR were shown in Supplementary Note 2. We selected the photosensitive resin (DSM IMAGE8000) to make the spiral bases by 3D printing. We selected the commonly used piezoelectric ceramic transducer (PZT) to fabricate the piezoelectric stack. We chose standardized metal copper to make copper pillars. The details of material property study in MMR were shown in Supplementary Note 3 and Supplementary Fig. 2.

## Characterization of micro-motion stimulation on MMR

We characterized the micro-motion stimulation by detecting micro-motions under different intensities, directions, frequency range and SNRs. For the detection limit, we demonstrated that the lowest detection limit was estimated to reach $10^{-6}$ g in the order of magnitude which was shown in Supplementary Note 8. For the micro-motion range, we also demonstrated that the direction range of micro-motion was $d_{33}$ direction which was along the $z$-direction in Cartesian coordinates. The frequency range of micro-motion could be 0–12 kHz by customizing the design of MMR. Moreover, through comparative experiments as shown in Supplementary Note 13 and Supplementary Figs. 16 and 17, we verified that MMR was able to detect micro-motions with SNR of −20 dB, which could be combined with deep learning to have a nearly 100% recognition accuracy under 200 training samples.

## Analytical derivations and numerical simulations

We explained the mechanical frequency-division multiplexing mechanism of the MMR with analytical derivations and numerical simulations. First, we performed the detailed derivations in MATLAB for the theoretical analysis. We derived the frequency-dependent effective mass and effective piezoelectric coefficient of a unit cell in Supplementary Note 1. The theoretical derivation of effective properties of the mechanical frequency-division multiplexing system was presented in Supplementary Note 5, and the corresponding results were plotted in Supplementary Fig. 5. We presented the parameter details during the theoretical derivation in Supplementary Table 2.

Then, we carried out the finite element simulation in COMSOL Multiphysics 5.4 for the numerical simulations. In the simulation analysis, we set the material parameters of photosensitive resin, piezoelectric stack, and copper pillar as follows: $E_{res} = 2.5$ Gpa, $E_{pie} = 117.4$ Gpa, and $E_{cop} = 110.0$ Gpa; shear modulus $G_{res} = 1.025$ Gpa, $G_{pie} = 23.5$ Gpa, and $G_{cop} = 38.5$ Gpa; mass density $\rho_{res} = 1250$ Kg m$^{-3}$, $\rho_{pie} = 7500$ Kg m$^{-3}$, and $\rho_{cop} = 8960$ Kg m$^{-3}$. The piezoelectric tensor $d_{33}$ and dielectric constant of the piezoelectric stack were set to 554 pC N$^{-1}$ and 1433.6, respectively. For the dispersion curve simulation as shown in Fig. 2c, we calculated the numerical bandgap in the Γ-M direction of the unit cell ($\theta_n = 11.25$) with the Bloch boundary conditions. For the simulation of frequency responses, we applied a swept micro-motion excitation $ACC(\omega)$ to MMR and obtained the voltage output $VOL_n(\omega)$ across the piezoelectric stack in the $n$th unit cell. We calculated the $d_{33eff}^n(\omega)$ by following the formula $d_{33eff}^n(\omega) = C_0|VOL_n(\omega)/ACC(\omega)|$ in the numerical results, where $C_0$ is the capacitance of the piezoelectric stack.

## Experimental setup and signal measurements

For the experimental $d_{33eff}^n(\omega)$ as shown in Fig. 2f, we first constructed a swept harmonic signal through a signal generator (SDG2122X, SIGLENT). Then, the constructed signal was amplified by a power amplifier (YE5872A, Sinocera Piezotronics INC) to drive the exciter (B&K4808, HBK) for generating micro-motion excitation $a(t)$. We fixed the MMR to the exciter, and the micro-motion excitation activated electric displacement of the unit cell in the $z$-direction. We also fixed the conventional piezoelectric mechanoreceptor (B&K4371, HBK) on the exciter as a comparison. We extracted the piezoelectric voltage $v_n(t)$ of each measurement channel through the data acquisition equipment (DH8300N, Donghua Jiaozhun). We further took the Fourier transform to convert the time-domain signals $a(t)$ and $v_n(t)$ to frequency-domain signals $A(\omega)$ and $V_n(\omega)$. Last, we obtained the $d_{33eff}^n(\omega)$ of each measurement channel by calculating the formula $d_{33eff}^n(\omega) = C_0|Vn(\omega)/A(\omega)|$ in the experimental results, where $C_0$ is the capacitance of the employed piezoelectric stack in MMR.

For the performance evaluation experiments, we exploited the above experimental equipment to build a micro-motion sensing platform as shown in Fig. 3a. We presented the details of experimental setup in Supplementary Fig. 7. During the signal measurement process, we employed a reconstruction algorithm to achieve the computational multi-channel demodulation of the measured signal, leading to a computational signal reconstruction approach. The detail of computational multi-channel demodulation was described in Supplementary Note 6 and Supplementary Fig. 8. We constructed an intelligent sensing system as shown in Fig. 4, in which we combined the in-situ signal enhancement of MMR and a deep neural network to obtain high recognition accuracy. The details of the deep neural network in this intelligent sensing system were presented in Supplementary Note 11 and Supplementary Fig. 13. We demonstrated that MMR could detect three movement states of the rotor test bench during the start-up process as shown in Supplementary Note 17 and Supplementary Fig. 25. We also demonstrated that MMR exhibited strong stability and reliability in different application scenarios as shown in Supplementary Note 18 and Supplementary Fig. 26.

## Data availability

All data needed to evaluate the conclusions in the paper are present in the paper and/or the Supplementary Information. Additional data related to this paper may be requested from the authors.

## Code availability

Source code and processed data are available from the corresponding author upon reasonable request.

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

## Acknowledgements

This work was supported by the National Natural Science Foundation of China (Grant No: 52275116, 11872244, and 12121002), the Program of Shanghai Academic/Technology Research Leader (Grant No: 22XD1421700) and the National Program for Support of Top-Notch Young Professionals. Q.H. and C.L. thanks Dr. Huachen Cui from the University of California, Los Angeles for his great suggestions on this manuscript.

## Author contributions

Q.H. and C.L. conceived the concepts and designed the research. C.L. and X.L. performed the fabrications, simulations, and experiments. C.L., Q.H. and X.L. drafted the manuscript. Q.H. supervised the entire project. Z.K.P. and G.M. supervised the theoretical study. All authors discussed the results, commented on, and revised the manuscript.

## Competing interests

The authors declare no competing interests.
