## [Peer Review File · Nature Communications]

Highly sensitive and broadband meta-mechanoreceptor via
mechanical frequency-division multiplexingReviewer #1 (Remarks to the Author):

Review for

Ultra-sensitive ultra-broadband meta-mechanoreceptor via mechanical frequency-division multiplexing

Comments to the Author :

In this manuscript, a metamaterial mechanoreceptor (MMR) is proposed which is comprised of piezoelectric resonators with distributed zero effective masses featuring a broad range of local resonances, leading to near-infinite sensitivity for micro-motion sensing within a broad bandwidth. The MMR is a promising candidate for ultra-sensitive ultra-broadband micro-motion sensing that was previously inaccessible for mechanics-guided mechanoreceptors. Therefore, I suggest it requires major revision for reconsideration. The detailed comments are as follows:

1. The authors keep saying that it remains a major challenge for mechanics-guided designs to simultaneously achieve high sensitivity and broadband sensing, so how much can be achieved by current research or products using other methods, such as bionics?
2. The structural parameters (h_1 , h_2 , h_3 , and a) should be systematically studied to determine the optimal size.
3. Why is the spiral base square at first (Fig 1, Fig 2a, and Fig 3a) and why is it round at the end (Fig 2e)? Is there an error here? Whether the shape affects the performance of the sensor?
4. The author made a rich application demonstration of MMR, and it would be better to explore its stability and reliability and give specific data.
5. A smart-driving assistance scene by equipping the MMR into an in-vehicle system is proposed. However, the operation of MMR needs the assistance of external equipment, such as a signal generator. For the automobile, the existence of this equipment is also a burden to the automobile itself. What is the significance of this?
6. In a practical application, there is a lot of noise in the motion of the car. How can the sensor eliminate this noise; otherwise, the application is impractical.
7. What is the anti-interference ability of the equipment to the outside world? Can it be applied to the industrial environments with more complex environments?

Reviewer #2 (Remarks to the Author):

The authors designed and fabricated a type of metamaterial mechanoreceptor inspired of rat vibrissae. The sensitivity of this mechanoreceptor is improved compared to conventional mechanics-guided mechanoreceptors. This design has bandwidth and ultrahigh sensitivity. Also, it is extendable towards both low and high frequency ranges through tuning the local resonance of each individual sensing cell. The authors approve their concept design of through theoretical analysis and experimental studies. The measurement system incorporating with artificial intelligence (AI) techniques will be achieves higher recognition accuracy. This paper could provide valuable concept to researchers who are interested in sensor design. the research work still needs to be enhanced and may additionally concepts and analysis be cleared before published. This reviewer recommends the publication after revision and review.

The reported metamaterial mechanoreceptor can sense micro-motion via piezoelectric materials under mechanical deformation. all design calculation, simulation and experimental is based on the motion in 33 direction. In the actual scenario, the unit cell, could be under different direction motion. Each unit cell is cantilever beam support a spiral base as rat vibrissae, which lateral motion is important. Also, for the rotation. For each K_n should be K_{n11} to K_{n66} . Please give the explanation and provement to approve this sensor is mono responded in one direction., since the many applications have complex motion.

Also, according to picture demonstration, the each responded signal of unit cell have been read by one circuit. With the unit cell increased, the connections affect may be morr serious. How to packaging them is important problem to be solved.

Reviewer's Comment:

In this manuscript, a metamaterial mechanoreceptor (MMR) is proposed which is comprised of piezoelectric resonators with distributed zero effective masses featuring a broad range of local resonances, leading to near-infinite sensitivity for micro-motion sensing within a broad bandwidth. The MMR is a promising candidate for ultra-sensitive ultra-broadband micro-motion sensing that was previously inaccessible for mechanics-guided mechanoreceptors. Therefore, I suggest it requires major revision for reconsideration.

Authors' Response:

We thank the reviewer for the positive and important comments on our work that helped us improve the quality of this manuscript. All the comments and suggestions have been carefully considered and the responses are provided as follows in detail. The revisions in the revised manuscript and supplementary information are marked in **BLUE** color.

Reviewer's Comment:

1. The authors keep saying that it remains a major challenge for mechanics-guided designs to simultaneously achieve high sensitivity and broadband sensing, so how much can be achieved by current research or products using other methods, such as bionics?

Authors' Response:

We thank the reviewer for the constructive comments on our work. We agree with the reviewers that further comparisons with other methods are needed in terms of sensitivity and bandwidth. In order to address this issue, we have further deeply thought about the main contributions of our work as compared to other methods. In the revisions, we have strengthened the significant contributions, summarized the significance and broad interests of our work.

For the design of artificial mechanoreceptors capable of micro-motion sensing, there's a trade-off between the sensitivity and bandwidth of mechanics-guided designs due to the nature of resonance effect. To address this challenge, we have reported a bioinspired MMR for ultra-sensitive ultra-broadband micro-motion sensing via mechanical frequency-division multiplexing in the revised manuscript. MMR addresses the trade-off between sensitivity and bandwidth in conventional mechanics-guided micro-motion sensors. This is achieved by tessellating piezoelectric resonators with distributed zero effective masses into a metamaterial architecture that performs as MMR for micro-motion sensing. MMR exhibits frequency-dependent effective piezoelectric coefficients that break the limits of intrinsic piezoelectric materials in which the electromechanical conversion is described by existing piezoelectric tensors under quasi-static stations. Experiments confirmed that the maximum sensitivity of MMR was improved by two orders of magnitude compared with the conventional piezoelectric mechanoreceptor, and its bandwidth with ultrahigh sensitivity was extendable towards both low-frequency and high-frequency ranges in 0-12 kHz. This manuscript demonstrates three applications based on MMR, including spatio-temporal sensing, remote-vibration monitoring and smart-driving assistance, and also explores structural health monitoring in industrial scenarios. Besides the applications demonstrated in this manuscript, our work revealed that MMR may be a promising candidate for ultra-sensitive ultra-broadband micro-motion sensing in mechanics-guided sensors, providing a new perspective on simpler mechanics-guided designs of high-performance sensors for various other physical information.

The main contribution of our work focuses on the design principle of the mechanoreceptor: The MMR is comprised of piezoelectric resonators with distributed zero effective masses featuring a broad range of local resonances, leading to near-infinite sensitivity for micro-motion sensing within a broad bandwidth. The MMR combines unit cells with mechanics-guided piezoelectric resonator design to bypass the limitations of intrinsic piezoelectric materials and achieve high sensitivity. The conventional piezoelectric coefficient d_{33} defined under quasi-static constraints is not sufficient to describe the piezoelectric property of the unit cell with zero effective mass under dynamic frequency-varying conditions. We define the frequency-dependent effective piezoelectric coefficient $d_{33\text{eff}}(\omega)$ based upon elastodynamics theory to describe the electromechanical conversion of the unit cell in different frequencies. Generally, the sensitivity increases with the increase of $d_{33\text{eff}}(\omega)$. We have demonstrated that the maximum $d_{33\text{eff}}(\omega)$ reached an appreciable value of 24930 pC N^{-1} , which was much higher than that of the state-of-art piezoelectric materials. The comparison of effective piezoelectric coefficient between the unit cell under dynamic frequency-varying conditions in this study and typical piezoelectric materials under quasi-static conditions has been revised in Fig. 2d of the main text as follows:

“

Fig. 2d, Comparison of effective piezoelectric coefficient between the unit cell under dynamic frequency-varying conditions in this study and typical piezoelectric materials under quasi-static conditions

”

In addition, the major points discussed above have been included in the revised manuscript addressing this issue as follows:

“Through analytical, numerical and experimental verifications, we demonstrate that these piezoelectric resonators with zero effective masses exhibit frequency-dependent effective piezoelectric coefficients, which are not achievable in intrinsic piezoelectric materials and lead to near-infinite sensitivity within a broad bandwidth. We tessellated these piezoelectric resonators as unit cells into a metamaterial architecture that performs as “meta-mechanoreceptor” for micro-motion sensing. We developed a mechanical frequency-division multiplexing system to mechanically modulate micro-motion signals in non-overlapping frequency bands, which combined a computational multi-channel demodulation approach to achieve the signal reconstruction. Compared with the conventional piezoelectric mechanoreceptors with a sensitivity of $0.812 \text{ mv m}^{-1}\text{s}^2$, MMR has a maximum sensitivity that is 2 orders of magnitude higher, reaching up to $36.540 \text{ mv m}^{-1}\text{s}^2$. Through integrating sensing units with prescribed piezoelectric coefficients, the bandwidth of MMR can be extended towards both low frequencies and high frequencies in 0-

12 kHz, leading to broad applications in spatio-temporal sensing, remote-vibration monitoring and smart-driving assistance. The designed MMR is a promising candidate for ultra-sensitive ultra-broadband micro-motion sensing that has not been attainable in conventional mechanics-guided micro-motion sensors, breaking the trade-off between sensitivity and bandwidth in the state-of-art mechanoreceptors with single-piece piezoelectric elements.” (Page 4, Line 85)

“When the unit cell was excited by external micro-motions (finite F) at the zero-mass frequency, the dynamic response was enhanced by the zero effective mass ($m_{\text{eff}}(\omega) \rightarrow 0$) with a near-infinite acceleration amplitude ($F/m_{\text{eff}}(\omega) \rightarrow \infty$). We applied micro-motion excitation to the unit cell in different directions, and confirmed that the unit cell was mono responded in d_{33} direction (Supplementary Note 3). We calculated the $m_{\text{eff}}(\omega)$ and $d_{33\text{eff}}(\omega)$ of the unit cell with the numerical method, which showed excellent agreement with the analytical results (see the red area in Fig. 2b). Due to the inevitable damping in experiments, the maximum $d_{33\text{eff}}(\omega)$ reached an appreciable value of 24930 pC N⁻¹ instead of a theoretical near-infinite value (Fig. 2d), which was much higher than that of the state-of-art piezoelectric materials⁴⁹⁻⁵⁷. Therefore, configuring zero effective mass design in the unit cell results in a near-infinite effective piezoelectric coefficient, allowing access to micro-motion sensing with an ultrahigh sensitivity around the zero-mass frequency.” (Page 7, Line 149)

- 49 Qiu, C. *et al.* Transparent ferroelectric crystals with ultrahigh piezoelectricity. *Nature* **577**, 350-354, doi:10.1038/s41586-019-1891-y (2020).
- 50 Wang, D. *et al.* Ultrahigh piezoelectricity in lead-free piezoceramics by synergistic design. *Nano Energy* **76**, doi:10.1016/j.nanoen.2020.104944 (2020).
- 51 Bijalwan, V. *et al.* Processing of 0.55(Ba_{0.9}Ca_{0.1})TiO₃-0.45Ba(Sn_{0.2}Ti_{0.8})O₃ lead-free ceramics with high piezoelectricity. *Journal of the American Ceramic Society* **103**, 4611-4624, doi:10.1111/jace.17090 (2020).
- 52 Chandrakala, E., Praveen, J. P. & Das, D. Enhanced Piezoelectricity in Lead-free BCZT Piezoceramics for Sensor Applications. *61st Dae-Solid State Physics Symposium* **1832**, doi:Artn 14003710.1063/1.4980819 (2017).
- 53 Wang, P., Li, Y. & Lu, Y. Enhanced piezoelectric properties of (Ba_{0.85}Ca_{0.15})(Ti_{0.9}Zr_{0.1})O₃ lead-free ceramics by optimizing calcination and sintering temperature. *Journal of the European Ceramic Society* **31**, 2005-2012, doi:10.1016/j.jeurceramsoc.2011.04.023 (2011).
- 54 Tao, H. *et al.* Ultrahigh Performance in Lead-Free Piezoceramics Utilizing a Relaxor Slush Polar State with Multiphase Coexistence. *J Am Chem Soc* **141**, 13987-13994, doi:10.1021/jacs.9b07188 (2019).
- 55 Liu, Q. *et al.* Practical high-performance lead-free piezoelectrics: structural flexibility beyond utilizing multiphase coexistence. *Natl Sci Rev* **7**, 355-365, doi:10.1093/nsr/nwz167 (2020).
- 56 Chen, M. *et al.* Polymorphic phase transition and enhanced piezoelectric properties in (Ba_{0.9}Ca_{0.1})(Ti_{1-x}Sn_x)O₃ lead-free ceramics. *Materials Letters* **97**, 86-89, doi:10.1016/j.matlet.2012.12.067 (2013).
- 57 Zuo, R. & Fu, J. Rhombohedral-Tetragonal Phase Coexistence and Piezoelectric Properties of (NaK)(NbSb)O₃-LiTaO₃-BaZrO₃ Lead-Free Ceramics. *Journal of the American Ceramic Society* **94**, 1467-1470, doi:10.1111/j.1551-2916.2010.04256.x (2011).

Moreover, we have compared our work with non-resonant methods, resonant methods, and bionic methods in detail from the perspective of sensitivity and bandwidth. We have demonstrated that MMR addressed the trade-off between sensitivity and bandwidth in conventional mechanics-guided micro-motion sensors. To clearly present the comparison of the sensing performance of MMR with other methods, we have added a new Supplementary Note 8 to the revised supplementary information as follows:

“We further compared our work with other methods in terms of sensitivity and bandwidth. **First**, the high sensitivity of the MMR is due to the frequency-dependent effective piezoelectric coefficient $d_{33\text{eff}}(\omega)$ generated by the zero effective masses. Here, d_{33} describes the correlation between the stress and the electric displacement in the intrinsic piezoelectric material along the z -direction in Cartesian coordinates. The relationship between the sensitivity $S(\omega)$ and $d_{33\text{eff}}(\omega)$ is:

$$S(\omega)=Cd_{33\text{eff}}(\omega) \quad (\text{S33})$$

where C is a positive constant coefficient. Generally, the sensitivity increases with the increase of $d_{33\text{eff}}(\omega)$, and we use $d_{33\text{eff}}(\omega)$ as an indicator to compare the sensitivity of different sensor designs. Here, we define $d_{33\text{eff}}(\omega)$ based upon elastodynamics theory to describe the electromechanical conversion of the unit cell in different frequencies as follows

$$d_{33\text{eff}}(\omega)=A(\omega)d_{33}\frac{1}{m_{\text{eff}}(\omega)} \quad (\text{S34})$$

where $A(\omega)$ represents a frequency-dependent variable related to the dynamic parameter of the unit cell, and ω denotes the varying frequency. As shown in **Fig. 2b** of the main text, we found that the $d_{33\text{eff}}(\omega)$ was theoretically a near-infinite value at the zero-mass frequency, which has not been demonstrated in existing piezoelectric sensors. As shown in **Fig. 2d** of the main text, some methods in Refs. [49-57] with non-resonant structural design are limited in sensitivity improvement. In conventional piezoelectric materials, the electromechanical conversion is described by existing piezoelectric tensors defined under quasi-static conditions. Due to the limitation of quasi-static conditions, most of the piezoelectric coefficients are about 700 pC N^{-1} , and only a few can reach 2100 pC N^{-1} as shown in Ref. [49]. While for MMR, the maximum $d_{33\text{eff}}(\omega)$ reached an appreciable value of 24930 pC N^{-1} , which was much higher than that of the state-of-art piezoelectric materials. The MMR combines unit cells with mechanics-guided piezoelectric resonator design to bypass the limitations of intrinsic piezoelectric materials and achieve high sensitivity. Essentially, configuring zero effective mass design in the unit cell results in a near-infinite effective piezoelectric coefficient, allowing access to micro-motion sensing with a high sensitivity around the zero-mass frequency.

Second, as shown in **Supplementary Fig. 11**, other methods improve sensitivity through resonant structural design, but limit the increase in bandwidth. **Supplementary Ref. [9]** developed a mode-localized resonant sensor with a bandwidth of 3.5 Hz^9 . **Supplementary Ref. [10]** and **Supplementary Ref. [11]** both developed a high resolution resonant sensor with a bandwidth of $5 \text{ Hz}^{10,11}$. **Supplementary Ref. [12]** developed a seismic-grade resonant sensor with a bandwidth of 100 Hz^{12} . **Supplementary Ref. [13]** developed a differential mode-localized sensor with a bandwidth of 320 Hz^{13} . **Supplementary Refs. [14-15]** developed a differential resonant sensor and a micromachined resonant sensor with a bandwidth of 500 Hz , respectively^{14,15}. **Supplementary Ref. [16]** developed a nanoresonator-based sensor with a bandwidth of 1000 Hz^{16} . **Supplementary Ref. [17]** developed a programmable resonant sensor with a bandwidth of 1246 Hz^{17} . **Supplementary Ref. [18]** developed a resonant accelerometer based on nanomechanical piezoresistive transduction with a bandwidth of 1500 Hz^{18} . The comparisons of bandwidth between other works with resonant structural design and our work are listed in **Supplementary Table 3**. It can be seen from the table that the bandwidth can reach 320 Hz through the mode-localized resonant structure design. Further, the bandwidth can reach 1.5 kHz through the structural design of the nano-resonator. While for our work, through integrating sensing units with prescribed piezoelectric coefficients, the bandwidth of MMR with high sensitivity can be extended towards both low frequencies and high frequencies in $0\text{-}12 \text{ kHz}$. Since MMR is a metamaterial structure composed of unit cells featuring a broad range of local resonances, the working bandwidth can be flexibly customized.

Third, as shown in **Supplementary Fig. 12**, there are also some bionic methods to realize acoustic sensing, vibration sensing and tactile sensing. **Supplementary Ref. [19]** developed a human cochlea-inspired acoustic sensor with the piezoelectric coefficient of 46 pC N^{-1} in a bandwidth of 1 kHz^{19} . **Supplementary Ref. [20]** developed a spider crack- inspired vibration sensor with the 100-fold sensitivity improvement in $0\text{-}1000 \text{ Hz}^{20}$. **Supplementary Ref. [21]** developed a human skin-inspired tactile sensor which can achieve a detection limit of 0.1 kPa with a bandwidth of 1 kHz^{21} . **Supplementary Ref. [22]** developed a chameleon skin-inspired vibration sensor, and its sensitivity was measured to be the output power under 0.245% strain with a

bandwidth of 5 Hz²². Other bionic methods focused on strain sensing and pressure sensing without consideration of the bandwidth. For example, **Supplementary Ref. [23]** developed a scorpion crack-inspired strain sensor, and its sensitivity was measured to be a Gauge factor of 1344.1 at 200% strain, which focused on the static strain sensing with no consideration of the bandwidth for dynamic micro-motion sensing²³. Similarly, **Supplementary Ref. [24]** developed a human skin-inspired strain sensing with the 24-fold sensitivity improvement, which also ignored the bandwidth measurement²⁴. Moreover, **Supplementary Refs. [25-28]** all developed the bio-inspired pressure sensors with the sensitivity of 8% Pa⁻¹, 50.17 kPa⁻¹, 8.5 kPa⁻¹ and 70.86% kPa⁻¹ respectively²⁵⁻²⁸, none of which considered the measurements of bandwidth. The comparisons of bandwidth between other bionic works and our work are listed in **Supplementary Table 4**. As can be seen from the table, the biomimetic approach has inspired different sensor designs, including: acoustic sensors, vibration sensors, tactile sensors, strain sensors, and pressure sensors. Different sensors have different definitions of sensitivity, and some consider bandwidth while others do not. On the whole, these bionic methods can achieve a piezoelectric coefficient of 46 pC N⁻¹, a 24-fold increase or a 100-fold increase in sensitivity, and some of their bandwidths can reach 1 kHz. For our work, MMR can achieve a piezoelectric coefficient of 24930 pC N⁻¹ and two orders of magnitude improvement in sensitivity. The MMR's bandwidth can reach 12 kHz, which can be extended towards both low frequencies and high frequencies.

In summary, we compare our work with non-resonant methods, resonant methods, and bionic methods in terms of sensitivity and bandwidth. We have reported a bioinspired MMR for ultra-sensitive ultra-broadband micro-motion sensing via mechanical frequency-division multiplexing. MMR addresses the trade-off between sensitivity and bandwidth in conventional mechanics-guided micro-motion sensors. This is achieved by tessellating piezoelectric resonators with distributed zero effective masses into a metamaterial architecture that performs as MMR for micro-motion sensing. This work can provide a new perspective on simpler mechanics-guided designs of high-performance sensors for various other physical information.” **(Supplementary Note 8)**

Supplementary Reference:

- [9] Pandit, M. *et al.* Closed-Loop Characterization of Noise and Stability in a Mode-Localized Resonant MEMS Sensor. *IEEE Trans Ultrason Ferroelectr Freq Control* **66**, 170-180, doi:10.1109/TUFFC.2018.2878241 (2019).
- [10] Pandit, M. *et al.* An Ultra-High Resolution Resonant Mems Accelerometer. *Proc Ieee Micr Elect*, 664-667, doi:10.1109/MEMSYS.2019.8870734 (2019).
- [11] Sobreviela-Falces, G. *et al.* A Mems Vibrating Beam Accelerometer for High Resolution Seismometry and Gravimetry. *2021 34th Ieee International Conference on Micro Electro Mechanical Systems (Mems 2021)*, 196-199, doi:10.1109/Mems51782.2021.9375431 (2021).
- [12] Xudong, Z., Thiruvengathanan, P. & Seshia, A. A. A Seismic-Grade Resonant MEMS Accelerometer. *Journal of Microelectromechanical Systems* **23**, 768-770, doi:10.1109/jmems.2014.2319196 (2014).
- [13] Pandit, M., Zhao, C., Sobreviela, G., Zou, X. & Seshia, A. A High Resolution Differential Mode-Localized MEMS Accelerometer. *Journal of Microelectromechanical Systems* **28**, 782-789, doi:10.1109/jmems.2019.2926651 (2019).
- [14] Shin, D. D. *et al.* Environmentally Robust Differential Resonant Accelerometer in a Wafer-Scale Encapsulation Process. *30th Ieee International Conference on Micro Electro Mechanical Systems (Mems 2017)*, 17-20, doi:10.1109/MEMSYS.2017.7863328 (2017).
- [15] Yin, Y., Fang, Z., Liu, Y. & Han, F. Temperature-Insensitive Structure Design of Micromachined Resonant Accelerometers. *Sensors (Basel)* **19**, doi:10.3390/s19071544 (2019).
- [16] Miani, T. *et al.* Nanoresonator-based accelerometer with large bandwidth and improved bias stability. *Int Symp Inert Senso*, doi:10.1109/Inertial53425.2022.9787526 (2022).
- [17] Xu, L., Wang, S., Jiang, Z. & Wei, X. Programmable synchronization enhanced MEMS resonant accelerometer. *Microsyst Nanoeng* **6**, 63, doi:10.1038/s41378-020-0170-2 (2020).
- [18] Miani, T. *et al.* Resonant Accelerometers Based on Nanomechanical Piezoresistive Transduction. *2021 34th Ieee International Conference on Micro Electro Mechanical Systems (Mems 2021)*, 192-195, doi:10.1109/Mems51782.2021.9375287 (2021).

- [19] Yan, W. *et al.* Single fibre enables acoustic fabrics via nanometre-scale vibrations. *Nature* **603**, 616-623, doi:10.1038/s41586-022-04476-9 (2022).
- [20] Fratzi, P. & Barth, F. G. Biomaterial systems for mechanosensing and actuation. *Nature* **462**, 442-448, doi:10.1038/nature08603 (2009).
- [21] Chun, S. *et al.* An artificial neural tactile sensing system. *Nature Electronics* **4**, 429-438, doi:10.1038/s41928-021-00585-x (2021).
- [22] Chen, X. *et al.* Bio-inspired flexible vibration visualization sensor based on piezo-electrochromic effect. *Journal of Materiomics* **6**, 643-650, doi:10.1016/j.jmat.2020.06.002 (2020).
- [23] Sun, H. *et al.* A Highly Sensitive and Stretchable Yarn Strain Sensor for Human Motion Tracking Utilizing a Wrinkle-Assisted Crack Structure. *ACS Appl Mater Interfaces* **11**, 36052-36062, doi:10.1021/acscami.9b09229 (2019).
- [24] Jiang, Y. *et al.* Auxetic Mechanical Metamaterials to Enhance Sensitivity of Stretchable Strain Sensors. *Adv Mater* **30**, e1706589, doi:10.1002/adma.201706589 (2018).
- [25] Takei, K. *et al.* Highly sensitive electronic whiskers based on patterned carbon nanotube and silver nanoparticle composite films. *Proc Natl Acad Sci U S A* **111**, 1703-1707, doi:10.1073/pnas.1317920111 (2014).
- [26] Su, B., Gong, S., Ma, Z., Yap, L. W. & Cheng, W. Mimosa-inspired design of a flexible pressure sensor with touch sensitivity. *Small* **11**, 1886-1891, doi:10.1002/sml.201403036 (2015).
- [27] Tang, H., Nie, P., Wang, R. & Sun, J. Piezoresistive electronic skin based on diverse bionic microstructure. *Sensors and Actuators A: Physical* **318**, doi:10.1016/j.sna.2020.112532 (2021).
- [28] Wang, J. *et al.* Bionic Fish-Scale Surface Structures Fabricated via Air/Water Interface for Flexible and Ultrasensitive Pressure Sensors. *ACS Appl Mater Interfaces* **10**, 30689-30697, doi:10.1021/acscami.8b08933 (2018).

In the revised supplementary information, we have plotted one new **Supplementary Fig. 11** to better present the comparison with other methods of resonant sensor design:

Supplementary Fig. 11 | Comparison with other methods of resonant sensor design. a, Mode-localized resonant sensor with a bandwidth of 3.5 Hz.⁹ **b**, Micro-machined resonant sensor with a bandwidth of 500 Hz.¹⁵ **c**, Nanoresonator-based resonant sensor with a bandwidth of 1000 Hz.¹⁶ **d**, Nanomechanical piezoresistive conduction with a bandwidth of 1500 Hz.¹⁸

In the revised supplementary information, we have also plotted one new **Supplementary Fig. 12** to better present the comparison with other methods of bionic sensor design:

Supplementary Fig. 12 | Comparison with other methods of bioinspired sensor design. **a**, Human cochlea-inspired acoustic sensor.¹⁹ **b**, Spider crack-inspired vibration sensor.²⁰ **c**, Human skin-inspired tactile sensor.²¹ **d**, Chameleon skin-inspired vibration sensor.²² **e**, Human skin-inspired strain sensor.²⁴ **f**, Mimosa-inspired pressure sensor.²⁷

In the revised supplementary information, we have also plotted one new **Supplementary Table 3** to better present the comparison details of bandwidth between the resonant sensors and our work:

Supplementary Table 3. Comparisons of bandwidth between the resonant sensors and our work

Methods	Ref.[9]	Ref.[10-11]	Ref.[12]	Ref.[13]	Ref.[14-15]	Ref.[16]	Ref.[17]	Ref.[18]	Our work
Band-width	3.5 Hz	5 Hz	100 Hz	320 Hz	500 Hz	1 kHz	1.2 kHz	1.5 kHz	12 kHz

In the revised supplementary information, we have also plotted one new **Supplementary Table 4** to better present the comparison details between the bioinspired sensors and our work:

Supplementary Table 4. Comparisons between the bioinspired sensors and our work

	Bio-inspired	Function	Sensitivity	Bandwidth
Ref.[19]	Human cochlea-inspired	Acoustic sensing	Piezoelectric coefficient of 46 pC N ⁻¹	1 kHz
Ref.[20]	Spider crack-Inspired	Vibration sensing	100-fold improvement	1 kHz
Ref.[21]	Human skin-inspired	Tactile sensing	Detection limit of 0.1 kPa	1 kHz
Ref.[22]	Chameleon skin-inspired	Vibration sensing	610 mW output power under 0.245% strain	5 Hz
Ref.[23]	Scorpion crack-Inspired	Strain sensing	Gauge factor of 1344.1 at 200% strain	No consideration
Ref.[24]	Human skin-inspired	Strain sensing	24-fold improvement	No consideration
Ref.[25]	Mammalian whisker-inspired	Pressure sensing	8% Pa ⁻¹	No consideration
Ref.[26]	Mimosa-Inspired	Pressure sensing	50.17 kPa ⁻¹	No consideration
Ref.[27]	Human skin-inspired	Pressure sensing	8.5 kPa ⁻¹	No consideration
Ref.[28]	Fish scale-Inspired	Pressure sensing	70.86% kPa ⁻¹	No consideration
Our work	Rat vibrissae-inspired	Micro-motion Sensing	Two orders of magnitude improvement (24930 pC N⁻¹)	12 kHz

Furthermore, in the revised manuscript, we have also revised the corresponding parts in the **Abstract**, **Introduction** and **Main text** as follows:

“We developed a mechanical frequency-division multiplexing mechanism for MMR, in which the measured micro-motion signal is mechanically modulated in non-overlapping frequency bands and reconstructed by a computational multi-channel demodulation approach. The maximum sensitivity of MMR is improved by two orders of magnitude compared to conventional mechanics-guided mechanoreceptors, and its bandwidth with ultrahigh sensitivity is extendable towards both low-frequency and high-frequency ranges in 0-12 kHz through tuning the local resonance of each individual sensing cell. The MMR is a promising candidate for ultra-sensitive ultra-broadband micro-motion sensing that was previously inaccessible for mechanics-guided mechanoreceptors,

opening new pathways towards spatio-temporal sensing, remote-vibration monitoring and smart-driving assistance.” (Page 2, Line 36)

“Nature has found a way to achieve remarkable micro-motion sensing by evolving biological mechanoreceptors¹. Biological mechanoreceptors, such as mouse vibrissae², fish lateral lines³, spider slits⁴, spider silk⁵, scorpion crack⁶ and human skin^{7,8}, are capable of detecting ubiquitous micro-motions with high sensitivity from a few hertz to thousands of hertz. Inspired by these naturally existing mechanoreceptors, artificial mechanoreceptors that convert mechanical micro-motions into electrical signals” (Page 2, Line 46)

2 Harrell, E. R., Goldin, M. A., Bathellier, B. & Shulz, D. E. An elaborate sweep-stick code in rat barrel cortex. *Sci Adv* **6**, doi:10.1126/sciadv.abb7189 (2020).

3 Montgomery, J. C., Baker, C. F. & Carton, A. G. The lateral line can mediate rheotaxis in fish. *Nature* **389**, 960-963, doi:10.1038/40135 (1997).

4 Barth, F. G. *A spider's world: senses and behavior*. (Springer Science & Business Media, 2002).

5 Zhou, J. & Miles, R. N. Sensing fluctuating airflow with spider silk. *Proc Natl Acad Sci U S A* **114**, 12120-12125, doi:10.1073/pnas.1710559114 (2017).

6 Sun, H. *et al.* A Highly Sensitive and Stretchable Yarn Strain Sensor for Human Motion Tracking Utilizing a Wrinkle-Assisted Crack Structure. *ACS Appl Mater Interfaces* **11**, 36052-36062, doi:10.1021/acsami.9b09229 (2019).

7 Tee, B. C. *et al.* A skin-inspired organic digital mechanoreceptor. *Science* **350**, 313-316, doi:10.1126/science.aaa9306 (2015).

8 Chun, S. *et al.* An artificial neural tactile sensing system. *Nature Electronics* **4**, 429-438, doi:10.1038/s41928-021-00585-x (2021).

“Moreover, when the SNR reaches -20 dB (Fig. 3e), the MMR can still measure the majority of the signal components, while the conventional piezoelectric mechanoreceptor cannot measure clear signal components since the chirp signal with SNR of -20 dB cannot reach the detection limit. We further demonstrated that MMR shows strong stability and reliability under long-time measurement of 1000 s and under the measurement of different SNRs from -30 dB to 10 dB (Supplementary Note 7). Compared to other methods (such as non-resonant^{49,50}, resonant^{59,60} and biomimetic^{61,62} designs), MMR addressed the trade-off between sensitivity and bandwidth, which was previously unachievable with the conventional piezoelectric mechanoreceptor (Supplementary Note 8), opening new opportunities for perception, monitoring and identification of environmental micro-motions.” (Page 10, Line 213)

Reviewer's Comment:

2. The structural parameters (h_1 , h_2 , h_3 , and a) should be systematically studied to determine the optimal size.

Authors' Response:

We thank the reviewer for the careful and constructive comments on our work. We agree with the reviewer that the structural parameters (h_1 , h_2 , h_3 , and a) should be systematically studied. Since the zero effective mass is the key to MMR design, we have systematically analyzed the effect of structural parameters on the zero effective mass in the revised manuscript. We designed the parameters of the unit cell as: $h_1 = 3.64$ mm, $h_2 = 5$ mm, $h_3 = 2.5$ mm and $a = 10$ mm. The range of the spiral angle is 8.0 to 17.5, which is to realize distributed zero effective masses at different frequencies. These structural parameters are determined through systematic research combined with the existing processing technology, which is close to the optimal design. To clearly present the details of structural parametric design in MMR, we have added a new Supplementary Note 2 to the revised supplementary information as follows:

“For the structural design of MMR, we determined the structural parameters through systematic parameter design combined with the existing fabrication technology. As shown in

Supplementary Figs. 1a-b, the structural parameters h_1 , h_2 , h_3 , and a of the unit cell are the height of the copper pillar, the height of the piezoelectric stack, the height of the spiral base, and the width of the spiral base, respectively. We configure the structural parameters to achieve the frequency-dependent zero effective mass. From the **Supplementary Note 1**, considering the condition of small damping, we set $m_{\text{eff}}(\omega)$ to 0 in **Supplementary Equation S9**, and the corresponding frequency ω_0 (defined as the zero-mass frequency) can be approximately described as:

$$\omega_0 = \sqrt{\frac{k_n}{m}} \cdot \sqrt{\frac{M+m}{M}} \quad (\text{S12})$$

Since the sensor is fixed to the surface of the object to be measured, the mass M will be much greater than the m . It can be seen from **Supplementary Equation S12** that the value of ω_0 mainly depends on k_n and m . In this design, ω_0 decreases as m increases, and ω_0 increases as k_n increases. Combined with the structure of the unit cell, h_1 and h_2 mainly determine the value of m . The value of h_3 has an influence on k_n and M . And a mainly determines the value of M . Combined with the above analytical derivation, we further use the numerical method to verify the influence of parameters.

We present the results of the parameter study in **Supplementary Figs. 1d-g**. We set the basic parameters of the unit cell as: $h_1 = 3.64$ mm, $h_2 = 5$ mm, $h_3 = 2.5$ mm, $a = 10$ mm and $\theta_n = 11.25$. First, we adjust h_1 from 1 mm to 10 mm while other parameters remain unchanged, and calculate the zero-mass frequency of the unit cell in the finite element analysis. From the numerical results as shown in **Supplementary Fig. 1d**, the ω_0 decreases with the increase of h_1 , which is caused by the increase of the m due to the increase of h_1 . Similarly, as shown in **Supplementary Fig. 1e**, we set h_2 from 1 mm to 10 mm, and also found that ω_0 decreases with the increase of h_2 . Since the density and cross-sectional area of the copper pillar are larger than those of the piezoelectric stack, the change of h_1 has a greater influence on ω_0 than the change of h_2 . Then, we set h_3 from 1 mm to 5 mm as shown in **Supplementary Fig. 1f**, and found that ω_0 increases with the increase of h_3 . This is because the increase in the thickness of the helical base leads to the increase in the thickness of the helical beam, which in turn increases the stiffness k_n , resulting in an increase in the ω_0 . Finally, we set a from 5 mm to 10 mm as shown in **Supplementary Fig. 1g**, and found that ω_0 changes slightly with the change of a . This is because an increase in a only causes an increase in M and no change in stiffness k_n . Combined with the above analysis of **Supplementary Equation S12**, the change of M has little effect on the change of ω_0 .

Essentially, the realization of zero-mass frequency ω_0 is the key to MMR design. We have systematically studied the effects of h_1 , h_2 , h_3 and a on the ω_0 . Our results are as follows: ω_0 decreases rapidly with the increase of h_1 ; ω_0 decreases slowly with the increase of h_2 ; ω_0 increases significantly with the increase of h_3 ; ω_0 approximately remains constant with the increase of a . Moreover, there is another parameter θ_n that has a greater impact on ω_0 . As shown in Fig. 2c of the main text, ω_0 decreases significantly as θ_n increases. For MMR, we design the structural parameters considering the fabrication method and cost. We first fabricated the spiral base of the unit cell by 3D printing photosensitive resin. In order to ensure the processing accuracy and strength of the spiral beam while keeping the size as small as possible, we design the parameters as $h_3 = 2.5$ mm and $a = 10$ mm. As shown in **Supplementary Fig. 1b**, the maximum range of θ_n can be 8.0 to 17.5 in a square base of 10 mm by 10 mm. Then, for copper pillars and piezoelectric stacks, we design parameters as follows in combination with current commercial machining and sintering processes: $h_1 = 3.64$ mm and $h_2 = 5$ mm. In such a structural design, the maximum sensitivity of MMR is improved by two orders of magnitude compared to conventional mechanics-guided mechanoreceptors, and its bandwidth with ultrahigh sensitivity is extendable towards both low-frequency and high-frequency ranges in 0-12 kHz through tuning the local resonance of each individual sensing cell.

In summary, we design the parameters of the unit cell as: $h_1 = 3.64$ mm, $h_2 = 5$ mm, $h_3 = 2.5$ mm and $a = 10$ mm. The range of the spiral angle is 8.0 to 17.5, which is to realize distributed zero effective masses at different frequencies. These structural parameters are determined through systematic study combined with the existing fabrication technology, which is close to the optimal design.” (Supplementary Note 2)

Additionally, in the revised supplementary information, we have plotted one new **Supplementary Fig. 1** to better present the frequency-dependent effective piezoelectric coefficient and parameter study:

Supplementary Fig. 1 | Frequency-dependent effective piezoelectric coefficient and parameter study. **a**, Single unit cell with applied excitation F in the z -direction. **b**, Spiral base of the unit cell. **c**, The simplified dynamical model of the unit cell with frequency-dependent zero effective mass. **d**, The effect of h_1 on ω_0 . **e**, The effect of h_2 on ω_0 . **f**, The effect of h_3 on ω_0 . **g**, The effect of a on ω_0 .

Moreover, we have also added the corresponding explanation in the **Methods** part of the revised manuscript to detail the structural parameter design as follows:

“Preparation of MMR began with the fabrication of unit cells. We determined the structural parameters through systematic parameter design combined with the existing processing technology. We first fabricated the spiral base of the unit cell with photosensitive resin by 3D printing technology. In order to ensure the fabrication accuracy and strength of the spiral beam while keeping the size as small as possible, we design the parameters as $h_3 = 2.5$ mm and $a = 10$ mm. The maximum range of θ_n can be 8.0 to 17.5 in a square base of 10 mm by 10 mm. Then we assembled the piezoelectric stack and the copper pillar to the spiral base as shown in Fig. 1d, which was to construct a piezoelectric resonator as a unit cell. For copper pillars and piezoelectric stacks, we design parameters as follows in combination with current commercial machining and sintering processes: $h_1 = 3.64$ mm and $h_2 = 5$ mm. The section of the piezoelectric stack is a square with a side length of 1.65 mm, and the section of the copper column is a circle with a radius of 2.2 mm. We fabricated unit cells with different θ_n and combined them into a scalable plane array as the MMR. In the experiments, the constructed MMR consisting of 3×3 unit cells was shown in Supplementary Fig. 6, and the parametric details are presented in Supplementary Table 1. The details of structural parametric design in MMR are shown in Supplementary Note 2.” (Page 14, Line 307)

Reviewer’s Comment:

3. Why is the spiral base square at first (Fig 1, Fig 2a, and Fig 3a) and why is it round at the end (Fig 2e)? Is there an error here? Whether the shape affects the performance of the sensor?

Authors’ Response:

We thank the reviewer for the careful comments on our work. The square spiral bases of unit cells in Fig 1, Fig 2a, and Fig 3a of the main text are structural schematic diagrams. As shown in **Figure R1a**, we tessellate nine unit cells with different θ_n into a plane metamaterial architecture for micro-motion sensing, which is denoted as MMR. To package the MMR, we added a ring outside the square base and encapsulated it in a circular housing as shown in **Figure R1b**. The fabricated MMR is shown in **Figure R1c**. Considering that the micro-motion information is transmitted to the nine unit cells through the outer ring during the measurement, we performed the simulation of the nine unit cells together with the outer ring in Fig. 2e of the main text. Overall, there are no errors in the original manuscript.

Additionally, to clearly explain whether the shape affects the performance of MMR, we have added the simulation of square bases as shown in Figure R1d. Combined with the analysis of the previous **Supplementary Fig. 1g**, the width of the spiral base has little effect on the zero-mass frequency. As shown in **Figure R1d**, the base with the outer ring differs slightly from the original square base in the width dimension. Since neither the spiral angle nor the thickness of the spiral beam changes, the base shape has little effect on the zero-mass frequency. We further simulated the zero-mass frequencies of round and square bases with finite element analysis, and we found that the nine zero-mass frequencies are consistent as shown in **Figure R1d**. We also calculated the dynamic effective piezoelectric coefficients $d_{33\text{eff}}^n(\omega)$ for round and square bases as shown in **Figure R1e**, and we found that $d_{33\text{eff}}^n(\omega)$ agrees well for both round and square bases. The results demonstrate that the shape of the spiral base does not affect the performance of the MMR. Moreover, since we should perform the calibration of MMR before the measurement and the actual calibration shall prevail, minor differences in dynamic properties would be corrected the base shape can be considered to have no influence.

Figure R1 | Effect of base shape. **a**, Structural details of MMR. **b**, Designed MMR. **c**, Fabricated MMR. **d**, Finite element analysis of round and square bases. **e**, Comparison of dynamic effective piezoelectric coefficient between the round and square bases.

In summary, there are no errors in the original manuscript. After comparative discussion and analysis, the shape of the base has no effect on the performance of the sensor.

Reviewer's Comment:

4. The author made a rich application demonstration of MMR, and it would be better to explore its stability and reliability and give specific data.

Authors' Response:

We thank the reviewer for the positive and constructive comments on our work. We also agree with the reviewer that the stability and reliability of MMR need to be explored. In the revised manuscript, we have verified the working performance of MMR through experiments under

different time and different signal-to-noise ratios. Experiments demonstrated that MMR shows strong stability and reliability under long-time measurement of 1000 s and under the measurement of different SNRs from -30 dB to 10 dB. Moreover, we also gave specific experimental data in demonstrated remote-vibration monitoring and smart-driving assistance, which confirmed that MMR showed strong stability and reliability in the measurement of amplitude stability with a zero drift of less than 0.01.

To clearly explain the stability and reliability of MMR, we have added a new Supplementary Note 7 to the revised supplementary information as follows:

“Since the MMR structure has strong stability below 65°C, the temperature change during the measurement at room temperature has little effect on the sensing performance. We mainly verified the stability and reliability of MMR through experiments under long-time measurement and under measurement of different SNRs. **First**, considering that the sensor needs to be calibrated before working, we study the amplitude stability of the sensor during a continuous long-time measurement. Since MMR is composed of multiple unit cells, we take the first unit cell (zero-mass frequency of 436 Hz) as an example to analyze its stability and reliability. As shown in Supplementary Fig. 9a, we firstly apply a swept-frequency micro-motion excitation to the MMR from 0 Hz to 3500 Hz at the room temperature of 25°C, and the excitation time of a single cycle is 10 seconds. From the output results (with normalized amplitude) of the unit cell, it can be seen that the unit cell has a strong output at about 1.25 s because the frequency of the excitation signal at this time is near the zero-mass frequency of the MMR. And the average value of the whole output signal does not deviate from the zero starting point, which means that the zero drift of the average amplitude is small. As shown in Supplementary Fig. 9b, to further quantify this zero drift of average amplitude, we applied a cyclic swept-frequency excitation to the MMR over 10 cycles with a duration of 100 s. We then calculated the zero drift of the MMR under 10 cycles of excitation as shown in Supplementary Fig. 9c. From the calculation results, the MMR has a slight zero drift with a periodic amplitude less than 0.1 under the long-time cyclic excitation. This may be due to the low-frequency disturbance oscillation caused by the sweep frequency excitation, which can be eliminated and calibrated by a simple filtering algorithm in the computational multi-channel demodulation process.

Second, we analyzed the amplitude stability of the unit cell when it is near resonance. We applied a 436 Hz harmonic excitation to the MMR and measured the amplitude output of the first unit cell. The details of the amplitude output are shown in Supplementary Fig. 9d. We measured the harmonic excitation continuously for 1000 s from the start operation with MMR at 25°C. The results of continuous measurement are shown in Supplementary Fig. 9e. Due to the unavoidable damping, MMR showed a stable resonant output, which resembled an enhanced forced oscillation with a stable output. When conducting experiments, we waited for the MMR to operate for about 20 s and started collecting data after the output is stable. We further calculated the maximum absolute value of the zero drift amplitude within 1000 seconds as shown in Supplementary Fig. 9f. From the calculated results, the zero drift is less than 0.01. This zero-drift can be corrected by the post-processing algorithm.

Third, we analyzed the stability and reliability of MMR at different signal-to-noise ratios (SNRs). As shown in **Supplementary Fig. 10**, we conducted comparative experiments with a chirp signal as the broadband signal with SNR of 10 dB, 5 dB, 0 dB, -10 dB, -20 dB and -30 dB, respectively. We used the MMR and the conventional piezoelectric mechanoreceptor to measure micro-motion signals under different signal-to-noise ratios and make comparative analysis. The conventional piezoelectric mechanoreceptor here is a piezoelectric accelerometer with non-resonant sensor design. We use the conventional piezoelectric mechanoreceptor as a measurement reference to analyze the measurement performance of MMR at different signal-to-noise ratios. From the measured results as shown in **Supplementary Figs. 10a-b**, when the SNR is 10 dB and 5 dB, both the MMR and the conventional piezoelectric mechanoreceptor can measure clear and similar time-varying frequency components. When the SNR is 0 dB, the measured result with MMR presented clearer time-varying frequency components than that with the conventional piezoelectric mechanoreceptor as shown in **Supplementary Fig. 10c**. When the SNR is -10 dB, MMR can still measure clear time-varying frequency components, but the measurement results from conventional piezoelectric mechanoreceptors show strong background noise as shown in **Supplementary Fig. 10d**. Moreover, when the SNR reaches -20 dB and -30 dB as shown in **Supplementary Figs. 10e-f**, the MMR can still measure the majority of the signal components, while the conventional piezoelectric mechanoreceptor cannot measure clear signal components since the chirp signal with SNR of -20 dB and -30 dB cannot reach the detection limit. As a result, under different SNRs, MMR is superior to conventional piezoelectric mechanoreceptors in micro-motion sensing, and has strong stability and reliability in sensing performance.

In summary, we further demonstrated that MMR shows strong stability and reliability under long-time measurement of 1000 s and under different SNRs from -30 dB to 10 dB.” (Supplementary Note 7)

In addition, to clearly explain the stability and reliability of MMR, we have also added the text in the main text as follows:

“Moreover, when the SNR reaches -20 dB (**Fig. 3e**), the MMR can still measure the majority of the signal components, while the conventional piezoelectric mechanoreceptor cannot measure clear signal components since the chirp signal with SNR of -20 dB cannot reach the detection limit. We further demonstrated that MMR shows strong stability and reliability under long-time measurement of 1000 s and under the measurement of different SNRs from -30 dB to 10 dB (Supplementary Note 7). Compared to other methods (such as non-resonant^{49,50}, resonant^{59,60} and biomimetic^{61,62} designs), MMR addressed the trade-off between sensitivity and bandwidth, which was previously unachievable with the conventional piezoelectric mechanoreceptor (Supplementary Note 8), opening new opportunities for perception, monitoring and identification of environmental micro-motions.” (Page 10, Line 213)

Moreover, in the revised supplementary information, we have plotted one new **Supplementary Fig. 9** to better present the stability and reliability under long-time measurement:

Supplementary Fig. 9 | Stability and reliability under long-time measurement. a, Swept-frequency signal under single cycle. **b**, Swept-frequency signal under ten cycles. **c**, Zero drift of measured amplitude under ten cycles. **d**, Measured amplitude of harmonic signal at 436.6 Hz. **e**, 1000-second continuous amplitude measurement of harmonic signals at 436.6 Hz. **f**, Zero drift of measured amplitude under continuous measurement of 1000 seconds.

In the revised supplementary information, we have also plotted one new **Supplementary Fig. 10** to better present the stability and reliability at different signal-to-noise ratios (SNRs):

Supplementary Fig. 10 | Stability and reliability at different signal-to-noise ratios (SNRs). **a**, Micro-motion sensing performance at a SNR of 10 dB. **b**, Micro-motion sensing performance at a SNR of 5 dB. **c**, Micro-motion sensing performance at a SNR of 0 dB. **d**, Micro-motion sensing performance at a SNR of -10 dB. **e**, Micro-motion sensing performance at a SNR of -20 dB. **f**, Micro-motion sensing performance at a SNR of -30 dB.

Furthermore, we gave specific experimental data in **Supplementary Fig. 23** to explain the stability and reliability in different application scenarios.

Supplementary Fig. 23 | Specific data on stability and reliability in different application scenarios. a, Musical scale signal and its **(b)** amplitude ability. **c**, Failed engine signal and its **(d)** amplitude ability. **e**, Failed bearing signal and its **(f)** amplitude ability.

Moreover, to further explain the stability and reliability in different application scenarios, we have added a new Supplementary Note 14 to the revised supplementary information as follows:

“Moreover, we also gave specific experimental data (with normalized amplitude) in demonstrated remote-vibration monitoring, smart-driving assistance and structural health monitoring as shown in **Supplementary Fig. 23**. From the results as shown in **Supplementary Fig. 23a**, we gave the specific data of 7 Do-Ti scales and 7 higher Do’-Ti’ scales in the demonstrated remote-vibration monitoring scenario. Each of the 14 scales lasts 0.5 seconds and is cycled twice. We then calculated the zero drift of the musical scale signal as shown in **Supplementary Fig. 23b**, from which we found that the magnitude of the zero drift is less than 0.01. More, we also gave the specific experimental data of failed engine signal in the demonstrated smart-driving assistance. As shown in **Supplementary Fig. 23c** and **Supplementary Fig. 23d**, we calculated that the zero drift of the normal engine signal is less than 0.01. Similarly, the zero drift of the failed bearing signal in the demonstrated structural health monitoring is also less than 0.01 as shown in **Supplementary Fig. 23e** and **Supplementary Fig. 23f**. In summary, we gave specific experimental data in demonstrated remote-vibration monitoring, smart-driving assistance and structural health monitoring, which demonstrated that MMR showed strong stability and reliability in the measurement of amplitude stability in different application scenarios.” (Supplementary Note 14)

Reviewer’s Comment:

5. A smart-driving assistance scene by equipping the MMR into an in-vehicle system is proposed. However, the operation of MMR needs the assistance of external equipment, such as a signal generator. For the automobile, the existence of this equipment is also a burden to the automobile itself. What is the significance of this?

Authors’ Response:

We thank the reviewer for the constructive comments on our work. In the demonstrated smart-driving assistance scene by equipping the MMR into an in-vehicle system, the operation of MMR does not need the assistance of a signal generator.

As shown in **Figure R2a**, we set up an experimental platform to evaluate the sensing performance of MMR. In these experimental setups, the signal from the signal generator is passed through the power amplifier to the vibration exciter. The vibration exciter then generates a micro-motion signal and applies it to the MMR, causing the MMR to output a corresponding electrical signal. During this process, the vibration exciter can be set by the computer to generate micro-motion sources of different frequencies and strengths to fully evaluate the performance of MMR. As shown in **Figure R2b**, in the demonstrated smart-driving assistance scene, neither the signal generator nor the vibration exciter is needed. This is because the engine itself generates vibration sources of different frequencies and strengths during operation. These micro-motions from the engine are exactly the signal sources that MMR needs to monitor for smart-driving assistance. As a result, in the demonstrated smart-driving assistance scene by equipping the MMR into an in-vehicle system, the operation of MMR does not need the assistance of a signal generator.

Although a signal generator is not required for the operation of MMR, a data collector and a data processor are still required. In our experiment, a data acquisition equipment was used to collect the electrical signal generated by MMR, and the micro-motion signal was reconstructed through a computational multi-channel demodulation algorithm in the computer. As shown in **Figure R2c**, we also provided some exploration possibilities that integrates the central processing unit (CPU) and the MMR. The CPU here integrates data acquisition, storage and calculation, which can be embedded in the MMR and connects the output of each unit cell to the corresponding acquisition channel as shown in **Figure R2d**. In this way, we can integrate MMR into an all-in-one sensing system as shown in **Figure R2e**, including sensing, storage and computing functions. This design can make the entire sensing system simpler and more portable.

Figure R2 | Integrated sensing of MMR. a, Photograph of the experimental setup for calibration. **b**, Smart-driving assistance. **c**, Central processing unit (CPU) and MMR. **d**, CPU integrating data collection, storage and calculation functions. **e**, Schematic diagram of integrated sensing with MMR.

In summary, experimental results demonstrated the proof-of-concept smart-driving assistance via MMR. In the demonstrated smart-driving assistance scene by equipping the MMR into an in-vehicle system, the operation of MMR does not need the assistance of a signal generator. The main contribution of our work focuses on the mechanics-guided design of micro-motion sensor, which may have more exploration possibilities in future research to improve the sensing performance. To further improve and broaden our research, we also provide an integrated sensing design for MMR with all-in-one sensing functions, which can make the whole sensing system more portable. For our current designed MMR, there are still many problems that need to be further explored when being applied in practical applications. This is expected to be solved in future research. We have also added corresponding exploration possibilities to the **Main text** addressing this issue as follows:

“By using advanced micro and flexible manufacturing technology^{66,67}, the width and height of MMR can be made in subwavelength, which is hoped to be integrated into various flexible micro-electro-mechanical systems. **By matching the integrated central processing unit (CPU), MMR can be packaged as an all-in-one system with sensing, storage and computing functions (Supplementary Note 15), which can enable the entire sensing system to be simpler and more portable⁶⁸⁻⁷⁰.** MMR can also provide high versatility in the Internet of Things by incorporating artificial intelligence (AI) techniques, which can be applied to sensing and computing in edge AI tasks⁷¹.” (Page 13, Line 298)

- 66 Skylar-Scott, M. A., Mueller, J., Visser, C. W. & Lewis, J. A. Voxellated soft matter via multimaterial multinozzle 3D printing. *Nature* **575**, 330-335, doi:10.1038/s41586-019-1736-8 (2019).
- 67 Zhang, C., McAdams, D. A., 2nd & Grunlan, J. C. Nano/Micro-Manufacturing of Bioinspired Materials: a Review of Methods to Mimic Natural Structures. *Adv Mater* **28**, 6292-6321, doi:10.1002/adma.201505555 (2016).
- 68 Bartolozzi, C. Neuromorphic circuits impart a sense of touch. *Science* **360**, 966-967, doi:10.1126/science.aat3125 (2018).
- 69 Zhang, Z. *et al.* All-in-one two-dimensional retinomorph hardware device for motion detection and recognition. *Nat Nanotechnol* **17**, 27-32, doi:10.1038/s41565-021-01003-1 (2022).
- 70 Dodda, A., Trainor, N., Redwing, J. M. & Das, S. All-in-one, bio-inspired, and low-power crypto engines for near-sensor security based on two-dimensional memtransistors. *Nat Commun* **13**, 3587, doi:10.1038/s41467-022-31148-z (2022).
- 71 Choi, C. *et al.* Reconfigurable heterogeneous integration using stackable chips with embedded artificial intelligence. *Nature Electronics* **5**, 386-393, doi:10.1038/s41928-022-00778-y (2022).

In the revised supplementary information, we also have added a new **Supplementary Note 15** to present future possibilities of integrated sensing of MMR as follows:

“In our experiment, a data acquisition equipment was used to collect the electrical signal generated by MMR, and the micro-motion signal was reconstructed through a computational multi-channel demodulation algorithm in the computer. As shown in **Supplementary Fig. 24a**, we also provided some exploration possibilities that integrates the central processing unit (CPU) and the MMR. Here the CPU integrates data acquisition, storage and calculation, which can be embedded in the MMR and connects the output of each unit cell to the corresponding acquisition channel as shown in **Supplementary Fig. 24b**. In this way, we can integrate MMR into an all-in-one sensing system as shown in **Supplementary Fig. 24c**, including sensing, storage and computing functions. This design can make the entire sensing system simpler and more portable.” (Supplementary Note 15)

Reviewer's Comment:

6. In a practical application, there is a lot of noise in the motion of the car. How can the sensor eliminate this noise; otherwise, the application is impractical.

Authors' Response:

We thank the reviewer for the careful and constructive comments on our work. The noise during the vehicle movement is mainly divided into two categories: one is the external sound noise in the cab, and the other is the internal background noise caused by the attenuation of the micro-motion signal by the vibration transmission path. Since the MMR has no related impedance matching design, most of the noise in the cab is reflected by the MMR and cannot be transmitted into it. Thus, the external sound noise in the cab interferes little with the operation of MMR. Here we mainly focus on the internal background noise caused by the attenuation of the micro-motion signal in the transmission process. As shown in the **Fig. 4e** of the main text in the revised manuscript, we identified the health status of engine by monitoring the micro-motions on the cab platform far away from the engine. Early abnormal vibration signal from the engine is attenuated during transmission, leading to low signal-to-noise ratio and great difficulty to be detected. For the working mechanism of MMR, there are two processes to improve the signal-to-noise ratio: one is the mechanics-guide design with zero effective mass that enhances the time-varying frequency components of the micro-motions, and the other is the adaptive filtering algorithm in the computational multi-channel demodulation that reduces the background noise.

Since the engine rotates periodically during the operation of the car, the relevant fault signals are mainly harmonic signals with time-varying frequency components. For the working mechanism of MMR, we have established a global dynamical model, and derived the effective piezoelectric coefficient $d_{33\text{eff}}^n(\omega)$ of the n th unit cell in this model as follows:

$$d_{33\text{eff}}^n(\omega) = A(\omega) d_{33} \frac{1}{M_{\text{eff}}(\omega)} \quad (\text{R1})$$

where $M_{\text{eff}}(\omega)$ is the frequency-dependent effective mass of MMR and generates distributed zero values in different frequencies. The MMR exhibits near-infinite $d_{33\text{eff}}^n(\omega)$ at these distributed zero-mass frequencies. This feature led to enhanced micro-motion responses when the frequencies of the fault signal components are around the zero-mass frequencies. From **Equation R1**, since the background noise is a signal with infinite bandwidth and the fault signal is a harmonic frequency component with a limited bandwidth, this zero-mass characteristic makes the enhancement of the fault signal much higher than that of the noise signal, which greatly improves the signal-to-noise ratio of the fault signal component.

Besides, we also use adaptive filtering algorithm to further improve the signal-to-noise ratio in the computational multi-channel demodulation. As shown in **Figure R3a**, with the mechanical frequency-division multiplexing mechanism in MMR, the engine vibration induced voltage output $v_n(t)$ was measured simultaneously through N unit cells. As shown in **Figure R3b**, due to the nonlinear modulation during measuring, we developed a computational multi-channel demodulation approach for signal reconstruction in **Supplementary Note 5**. We presented the detailed workflow of computational multi-channel demodulation as shown in **Supplementary Fig. 7**. During the working process of MMR, each unit cell was a narrowband measurement channel with piezoelectric voltage output, which modulated the micro-motion response nonlinearly in a specific frequency range. In **Supplementary Fig. 7**, we described how we reconstructed the unknown micro-motion excitation $s(t)$ from the measured data including the frequency responses $H_n(\omega)$ and piezoelectric voltage signals $v_n(t)$. From the workflow of the signal reconstruction process, we used a least-mean-square adaptive filter algorithm to process the measured voltage $v_n(t)$ of the n th channel and then performed Fourier transform algorithm to obtain the $V_n(\omega)$, which can further improve the signal-to-noise ratio.

Figure R3 | Sensing workflow for MMR. **a**, The schematic of micro-motion sensing system with MMR. **b**, Schematic of the computational multi-channel demodulation through solving the inverse problem. **c**, The constructed original signal S_1 consisting of nine harmonic components. **d**, At a signal-to-noise ratio (SNR) of -20 dB, the measured signal S_2 without MMR has strong background noise. **e**, The measured signal S_3 with MMR via computational multi-channel demodulation. **f**, The signal measured with MMR has a small error ($S_3 - S_1$) compared with the original signal even in a strong noise background.

Moreover, since the relevant fault signals from car engine are mainly harmonic signals with time-varying frequency components, we also presented the measurement details of time-vary frequency components. We constructed nine harmonic components of equal amplitude. The frequency values of nine harmonic components were around 437 Hz, 684 Hz, 964 Hz, 1250 Hz, 1540 Hz, 1900 Hz, 2130 Hz, 2480 Hz and 2880 Hz, respectively. We added background noise to the original multi-frequency signal to construct an experimental multi-frequency signal with an SNR of -20 dB. As shown in **Figure R3c**, each harmonic component appeared sequentially at 0.1 s intervals in the multi-frequency signal denoted by S_1 . The measured result that was denoted by S_2 without MMR was shown in **Figure R3d**, from which we found that the measured signal contained strong background noise. We presented the measured result denoted by S_3 with MMR in **Figure R3e**. We obtained the measurement error (S_3-S_1) from 0.45 s to 0.475 s as shown in **Figure R3f**. The average measurement error was less than 3%, which was calculated by the amplitude ratio of the error to the original signal. From the measured results and error analysis, we found that the measured results with MMR were relatively close to the original signal even in an SNR of -20 dB. For practical demonstration, we conducted experiments with six engine vibration signals, including one normal signal and five faulty signals as shown in **Fig. 4f** of the main text. The recognition accuracy with MMR was 95.3%, which was more than 5 times higher than the accuracy of 14.3% without MMR (**Supplementary Note 12**). Therefore, the demonstrated results revealed that MMR can improve the signal-to-noise ratio of the engine failure signal component.

In summary, for the working mechanism of MMR, there are two processes to improve the signal-to-noise ratio: one is the mechanics-guide design with zero effective mass that enhances the time-varying frequency components of the micro-motions, and the other is the adaptive filtering algorithm in the computational multi-channel demodulation that reduces the background noise. We have demonstrated a proof-of-concept smart-driving assistance, and MMR is expected to play an important role in the Internet of Vehicles system, serving for condition monitoring, fault diagnosis and intelligent maintenance. To better explain the mechanism of improving the signal-to-noise ratio, we have added further analysis in **Supplementary Note 12** addressing this issue as follows:

“ ...

The noise during the vehicle movement is mainly divided into two categories: one is the external sound noise in the cab, and the other is the internal background noise caused by the attenuation of the micro-motion signal by the vibration transmission path. Since the MMR has no related impedance matching design, most of the noise in the cab is reflected by the MMR and cannot be transmitted into it. Thus, the external sound noise in the cab interferes little with the operation of MMR. Here we mainly focus on the internal background noise caused by the attenuation of the micro-motion signal in the transmission process. As shown in the Fig. 4e of the main text in the revised manuscript, we identified the health status of engine by monitoring the micro-motions on the cab platform far away from the engine. Early abnormal vibration signal from the engine is attenuated during transmission, leading to low signal-to-noise ratio and great difficulty to be detected. For the working mechanism of MMR, there are two processes to improve the signal-to-noise ratio: one is the mechanics-guide design with zero effective mass that enhances the time-varying frequency components of the micro-motions, and the other is the adaptive filtering algorithm in the computational multi-channel demodulation that reduces the background noise.

“ ...
” (**Supplementary Note 12**)

Reviewer's Comment:

7. What is the anti-interference ability of the equipment to the outside world? Can it be applied to the industrial environments with more complex environments?

Authors' Response:

We thank the reviewer for the constructive comments on our work. We have demonstrated an application scene in smart-driving assistance as shown in **Fig. 4e** and **Fig. 4f** of the main text in the revised manuscript. The signals involved in the experiment are all real car engine failure signals from the outside world, which shows strong background noise. We have presented the signal details and measured results in **Supplementary Note 12**, **Supplementary Fig. 18**, **Supplementary Fig. 19** and **Supplementary Video 6**. We conducted experiments with six engine vibration signals, including one normal signal and five faulty signals. The recognition accuracy with MMR was 95.3%, which was more than 5 times higher than the accuracy of 14.3% without MMR. Since all signals are practical engine signals, the experimental results in smart-driving assistance have proved that MMR has sufficient anti-interference ability to the outside world. The demonstrated results revealed that MMR was a promising candidate for smart-driving assistance, and was expected to play an important role in the Internet of Vehicles system.

To further confirm the working ability of MMR and explore the possibility of application in industrial scenario, we illustrated that MMR enabled structural health monitoring in the helicopter transmission system by detecting early rotor failures, which was unachievable in the case without MMR. To clearly explain the structural health monitoring, we have added a new **Supplementary Note 13** to the revised supplementary information as follows:

“To further confirm the working ability of MMR and explore the possibility of application in industrial scenario, we added a demonstration of structural health monitoring as shown in **Supplementary Fig. 20**. Rotating machinery is the main body of mechanical equipment in aerospace and other industrial fields, and the research on fault diagnosis of rotating machinery is of great significance for the stable operation of equipment. **Supplementary Fig. 20a** shows a twin-engine multi-purpose helicopter which is a typical representative of rotating machinery in aerospace. Its transmission system includes main rotor shaft, main reducer, tail transmission shaft, connecting shaft and tail reducer as shown in **Supplementary Fig. 20b**. The tail transmission shaft is a slender shaft located between the main reducer and the tail reducer, on which four bearings of identical type are arranged as support and transmission. Due to the limitation of sensor measurement arrangement location, identifying weak faults of remote bearings is crucial to its structural health monitoring.

For the helicopter transmission system, we built a rotor test bench for structural health monitoring of slender tail shaft, which is a more complex application environment due to the variety and unpredictability of early bearing failures. As shown in **Supplementary Fig. 20c**, we

arrange MMR on the top of the right end bearing seat, and then replace the bearing inside the left end seat housing with an early weak failure bearing. As shown in **Supplementary Fig. 20d**, the remote bearing failure is sensed by the MMR and reconstructed to obtain the signal $s(t)$, which is further combined with the deep neural network (DNN) to realize the Monitoring and identification of early weak faults. As shown in **Supplementary Fig. 21**, we present the signal measurement details of early weak faults. We first machined three bearings with early failure, including inner race faults (Fault #1), outer race faults (Fault #2) and rolling element faults (Fault #3). Next, experiments were carried out under the normal state (Normal) and three faulty states, respectively. It can be seen that the test results based on MMR show clear time-frequency representations relatively, and the time-varying frequency components corresponding to each faulty state are different. In contrast, the four signals measured by the conventional piezoelectric accelerometer under the normal state and three fault states are similar. The results show that the early faults generated by the left end bearing are greatly attenuated when transmitted to the right end bearing seat, which is difficult for the conventional piezoelectric accelerometer to measure.

Then, we input the time-frequency representations with and without MMR into the constructed deep neural network, respectively. Here, we constructed 400 signal samples, 50% of which were used for training and the other 50% for recognition. We presented the recognition details as shown in **Supplementary Fig. 22**. From the recognition results with MMR, the overall identification accuracy was 96% shown in **Supplementary Fig. 22a**. In contrast, from the recognition results without MMR, most samples were identified incorrectly and the average identification accuracy rate was 27%, which was far lower than the recognition performance with MMR as shown in **Supplementary Fig. 22b**. The high recognition accuracy in the demonstrated smart-driving assistance was attributed to the ultra-sensitive ultra-broadband property of MMR.”
(Supplementary Note 13)

In summary, since all signals demonstrated in the smart-driving assistance scene are real engine signals, the experimental results have proved that MMR has sufficient anti-interference ability when identifying abnormal engine vibrations. Besides, we further add a demonstration of structural health monitoring for slender tail shaft of helicopter. Experimental results show that MMR still exhibit sensing performance unattainable with conventional piezoelectric accelerometers, which proves that MMR can have great potential in industrial application scenarios. Both application demonstrations prove that MMR has sufficient anti-interference ability when applied in complex environments.

In the revised supplementary information, we have plotted one new **Supplementary Fig. 20** to better present the structural health monitoring in the helicopter transmission system as follows:

Supplementary Fig. 20 | Demonstration of structural health monitoring. a, Helicopter. **b**, Helicopter drive system with slender tail shaft. **c**, Rotor test bench for structural health monitoring of slender tail shaft. **d**, Schematic diagram of the experimental layout.

In the revised supplementary information, we have also plotted one new **Supplementary Fig. 21** to better present the details of signal measurements in structural health monitoring:

Supplementary Fig. 21 | Details of signal measurements in structural health monitoring. The measured vibration signals from the bear, including one normal signal and three fault signals (Fault #1-#3). The signal measurements corresponding to (a) “Normal”, (b) “Fault #1”, (c) “Fault #2” and (d) “Fault #3” with and without MMR.

In the revised supplementary information, we have also plotted one new **Supplementary Fig. 22** to better present the recognition comparison with and without MMR in structural health monitoring:

Supplementary Fig. 22 | Recognition comparison with and without MMR in structural health monitoring. a, The average recognition accuracy with MMR is 96%. **b,** The average recognition accuracy without MMR is 27%.

We would like to thank the reviewer for the careful and constructive review on our work again. In the revised manuscript and supplementary information, we have carefully considered the reviewers' suggestions and comments and provided corresponding revisions, hoping that these revisions are satisfactory.

Reviewer's Comment:

The authors designed and fabricated a type of metamaterial mechanoreceptor inspired of rat vibrissae. The sensitivity of this mechanoreceptor is improved compared to conventional mechanics-guided mechanoreceptors. This design has bandwidth and ultrahigh sensitivity. Also, it is extendable towards both low and high frequency ranges through tuning the local resonance of each individual sensing cell. The authors approve their concept design of through theoretical analysis and experimental studies. The measurement system incorporating with artificial intelligence (AI) techniques will be achieves higher recognition accuracy. This paper could provide valuable concept to researchers who are interested in sensor design. the research work still needs to be enhanced and may additionally concepts and analysis be cleared before published. This reviewer recommends the publication after revision and review.

Authors' Response:

We thank the reviewer for the positive and important comments on our work that helped us improve the quality of this manuscript. All the comments and suggestions have been carefully considered and the responses are provided as follows in detail. The revisions in the revised manuscript and supplementary information are marked in **BLUE** color.

Reviewer's Comment:

1. The reported metamaterial mechanoreceptor can sense micro-motion via piezoelectric materials under mechanical deformation. all design calculation, simulation and experimental is based on the motion in 33 direction. In the actual scenario, the unit cell, could be under different direction motion. Each unit cell is cantilever beam support a spiral base as rat vibrissae, which lateral motion is important. Also, for the rotation. For each Kn should be Kn11 to Kn66. Please give the explanation and provement to approve this sensor is mono responded in one direction., since the many applications have complex motion.

Authors' Response:

We thank the reviewer for the careful and constructive comments on our work. We agree with the reviewer that this sensor needs to be explained and demonstrated to be mono responded in one direction. In order to unify the symbols, we use d_{11}^n to d_{66}^n to represent Kn11 to Kn66 mentioned by reviewer. In the revised manuscript, we have added the explanation and proof to approve MMR is mono responded in d_{33} direction.

To clearly explain the details of the mono response of MMR in d_{33} direction, we have added a new **Supplementary Note 3** to the revised supplementary information as follows:

“Since the practical micro-motions are multi-directional, we further give the explanation to approve MMR is mono responded in d_{33} direction as shown in **Supplementary Fig. 2**. Here, d_{33} describes the correlation between the stress and the electric displacement in the intrinsic piezoelectric material along the z -direction in Cartesian coordinates. As shown in **Supplementary Fig. 2a**, the piezoelectric stack is between the spiral base and the copper pillar, and converts the micro-motion excitation from the spiral base into a voltage signal. The piezoelectric stack adopts

the Z-direction stacking manufacturing process, and we have tested that only the Z-direction charge output is generated. Since the piezoelectric stack is installed on the helical base, applying weak vibration excitations in different directions to the helical base can only cause the piezoelectric stack to produce voltage outputs in the Z direction. Here, the micro-motions in different directions can cause the stress or shear force in different directions in the spiral base. As shown in **Supplementary Fig. 2b**, we list all 6 possible micro-motion directions applied to the spiral base, including stress direction along X (denoted as 1), stress direction along Y (denoted as 2), stress direction along Z (denoted as 3), shear force direction around X rotation (denoted as 4), shear force direction around Y rotation (denoted as 5) and shear force direction around Z rotation (denoted as 6). Since the n th unit cell only has voltage output in the Z direction (denoted as 3), only six directions between d_{11}^n and d_{66}^n are valid for MMR, and the six directions are d_{31}^n , d_{32}^n , d_{33}^n , d_{34}^n , d_{35}^n and d_{36}^n , respectively. We present the numerical simulations in six different directions with finite element analysis method as shown in **Supplementary Figs. 2c-h**, respectively. We apply swept-frequency micro-motion excitations of equal amplitude and different directions to the spiral base, and then calculate the voltage output of the unit cell in the Z direction.

As shown in **Supplementary Fig. 3**, we present voltage output details for different directions and normalize the output amplitude of the voltage. Taking the d_{33}^n direction as the normalized benchmark, the maximum output amplitude value of d_{33}^n is 1, while the maximum output amplitude values in the d_{31}^n , d_{32}^n , d_{34}^n , d_{35}^n and d_{36}^n directions are 2.2e-3, 2.05e-3, 1.4e-3, 1.8e-3 and 8e-3, respectively. From the results, the voltage output in the other five directions is far smaller than the voltage output in the direction of d_{33}^n , which can be approximated that MMR is mono responded in d_{33}^n direction.” (Supplementary Note 3)

In addition, the major points discussed above have been included in the revised manuscript addressing this issue as follows:

“When the unit cell was excited by external micro-motions (finite F) at the zero-mass frequency, the dynamic response was enhanced by the zero effective mass ($m_{\text{eff}}(\omega) \rightarrow 0$) with a near-infinite acceleration amplitude ($F/m_{\text{eff}}(\omega) \rightarrow \infty$). We applied micro-motion excitation to the unit cell in different directions, and confirmed that the unit cell was mono responded in d_{33} direction (Supplementary Note 3). We calculated the $m_{\text{eff}}(\omega)$ and $d_{33\text{eff}}(\omega)$ of the unit cell with the numerical method, which showed excellent agreement with the analytical results (see the red area in **Fig. 2b**). Due to the inevitable damping in experiments, the maximum $d_{33\text{eff}}(\omega)$ reached an appreciable value of 24930 pC N⁻¹ instead of a theoretical near-infinite value (**Fig. 2d**), which was much higher than that of the state-of-art piezoelectric materials⁴⁹⁻⁵⁷. Therefore, configuring zero effective mass design in the unit cell results in a near-infinite effective piezoelectric coefficient, allowing access to micro-motion sensing with an ultrahigh sensitivity around the zero-mass frequency.” (Page 7, Line 149)

In summary, we have added data and explanations to demonstrate that MMR can be responded in six directions, include d_{31}^n , d_{32}^n , d_{33}^n , d_{34}^n , d_{35}^n and d_{36}^n . Since the output of the d_{33} direction is far greater than the other five directions with very small output, it can be approximated that MMR is mono responded in d_{33} direction.

In the revised supplementary information, we have plotted one new **Supplementary Fig. 2** to better present the response simulation of unit cell under excitation in different directions as follows:

Supplementary Fig. 2 | Response simulation of unit cell under excitation in different directions. **a**, The unit cell model. **b**, Six possible directions of micro-motion excitation on the spiral base. **c**, Simulation in six different directions corresponding to (c) “ d_{31}^n ”, (d) “ d_{32}^n ”, (e) “ d_{33}^n ”, (f) “ d_{34}^n ”, (g) “ d_{35}^n ” and (h) “ d_{36}^n ”.

In the revised supplementary information, we have also plotted one new **Supplementary Fig. 3** to better present the response details of unit cell in different directions as follows:

Supplementary Fig. 3 | Response details of unit cell in different directions. The voltage output amplitude of the unit cell in six directions corresponds to (a) “ d_{31} ”, (b) “ d_{32} ”, (c) “ d_{33} ”, (d) “ d_{34} ”, (e) “ d_{35} ” and (f) “ d_{36} ” under equal-amplitude swept-frequency micro-motion excitation.

Reviewer's Comment:

2. Also, according to picture demonstration, the each responded signal of unit cell have been read by one circuit. With the unit cell increased, the connections affect may be more serious. How to packaging them is important problem to be solved.

Authors' Response:

We thank the reviewer for the careful and constructive comments on our work. We agree with the reviewer that the each responded signal of unit cell need to be read by one circuit. In our experiment, a data acquisition equipment was used to collect the electrical signal generated by MMR, and the micro-motion signal was reconstructed through a computational multi-channel demodulation algorithm in the computer. As shown in **Supplementary Fig. 24a**, we also provided some exploration possibilities that integrates the central processing unit (CPU) and the MMR. Here the CPU integrates data acquisition, storage and calculation, which can be embedded in the MMR and connects the output of each unit cell to the corresponding acquisition channel as shown in **Supplementary Fig. 24b**. In this way, we can integrate MMR into an all-in-one sensing system as shown in **Supplementary Fig. 24c**, including sensing, storage and computing functions. This design can make the entire sensing system simpler and more portable.

Supplementary Fig. 24 | Integrated sensing of MMR. a, Central processing unit (CPU) and MMR. **b**, CPU integrating data collection, storage and calculation functions. **c**, Schematic diagram of integrated sensing with MMR.

In summary, the main contribution of our work focuses on the mechanics-guided design of micro-motion sensor, which may have more exploration possibilities in future research to improve the sensing performance. To further improve and broaden our research, we also provide an integrated sensing design for MMR with all-in-one sensing functions, which can make the whole sensing system more portable. For our current designed MMR, there are still many problems that need to be further explored when being applied in practical applications. This is expected to be solved in future research. We have also added corresponding exploration possibilities to the **Main text** addressing this issue as follows:

“By using advanced micro and flexible manufacturing technology^{66,67}, the width and height of MMR can be made in subwavelength, which is hoped to be integrated into various flexible micro-electro-mechanical systems. By matching the integrated central processing unit (CPU),

MMR can be packaged as an all-in-one system with sensing, storage and computing functions (Supplementary Note 15), which can enable the entire sensing system to be simpler and more portable⁶⁸⁻⁷⁰. MMR can also provide high versatility in the Internet of Things by incorporating artificial intelligence (AI) techniques, which can be applied to sensing and computing in edge AI tasks⁷¹.” (Page 13, Line 298)

- 66 Skylar-Scott, M. A., Mueller, J., Visser, C. W. & Lewis, J. A. Voxelated soft matter via multimaterial multinozzle 3D printing. *Nature* **575**, 330-335, doi:10.1038/s41586-019-1736-8 (2019).
- 67 Zhang, C., McAdams, D. A., 2nd & Grunlan, J. C. Nano/Micro-Manufacturing of Bioinspired Materials: a Review of Methods to Mimic Natural Structures. *Adv Mater* **28**, 6292-6321, doi:10.1002/adma.201505555 (2016).
- 68 Bartolozzi, C. Neuromorphic circuits impart a sense of touch. *Science* **360**, 966-967, doi:10.1126/science.aat3125 (2018).
- 69 Zhang, Z. *et al.* All-in-one two-dimensional retinomorphic hardware device for motion detection and recognition. *Nat Nanotechnol* **17**, 27-32, doi:10.1038/s41565-021-01003-1 (2022).
- 70 Dodda, A., Trainor, N., Redwing, J. M. & Das, S. All-in-one, bio-inspired, and low-power crypto engines for near-sensor security based on two-dimensional memtransistors. *Nat Commun* **13**, 3587, doi:10.1038/s41467-022-31148-z (2022).
- 71 Choi, C. *et al.* Reconfigurable heterogeneous integration using stackable chips with embedded artificial intelligence. *Nature Electronics* **5**, 386-393, doi:10.1038/s41928-022-00778-y (2022).

In the revised supplementary information, we also have added a new Supplementary Note 15 to present future possibilities of integrated sensing of MMR as follows:

“In our experiment, a data acquisition equipment was used to collect the electrical signal generated by MMR, and the micro-motion signal was reconstructed through a computational multi-channel demodulation algorithm in the computer. As shown in Supplementary Fig. 24a, we also provided some exploration possibilities that integrates the central processing unit (CPU) and the MMR. Here the CPU integrates data acquisition, storage and calculation, which can be embedded in the MMR and connects the output of each unit cell to the corresponding acquisition channel as shown in Supplementary Fig. 24b. In this way, we can integrate MMR into an all-in-one sensing system as shown in Supplementary Fig. 24c, including sensing, storage and computing functions. This design can make the entire sensing system simpler and more portable.” (Supplementary Note 15)

We would like to thank you very much for the positive and constructive review on our work again and hope these revisions are satisfactory.

Reviewer #1 (Remarks to the Author):

In this revision, the authors provide explanations for the questions. But I recommend major revision as there are still some concerns listed below:

1. The micro-motion stimulation (vibration force/energy required for stimulation for this device) for this materials is not clearly characterized. This property is strongly dependent on the materials' mechanical properties and physical properties. The optimization of materials is not demonstrated.
2. The device used very small pixel and resolution, however, the motion stimulation would cause the whole device to vibrate and hard to distinguish the motion from various direction and position, if the force is parrallel to bottom base. Then the small pixel ad resolution would be meaningless. The authors need to define their limited applications.
3. There is no enough introduction of the conventional piezoelectric mechanoreceptor. Is this a commerical product? If it is, then they need to provide the details of the item, such as model/product number and manufacture name. If this is made by the authors, they need to introduce the details of the manufacturing processes, and compare with other products if possible. If the information of the conventional one is missing, it is hard to say their performance are superior to all other mechanoreceptors.
4. The detection limit and range of the motion is not demonstrated. The mechanical motions can be different with different styles of vibration and energy levels. The authors need to characterize all the details, and compare if possible.
5. The vibration and motions can be complex, while this device can convert one simple motion into more complex vibrations, such as rotation, translation. Then this device would not be able to detect the movement states, which making the device less functional.
6. For the detection of engine failure, the authors need to define the specific situations of failure detection. For sure, it would be very difficult to detect all engine troubles with this device. Firstly, the normal engines sometimes still can generate abnormal singals under bad road conditions, and they would have different signals under different days caused by weather, humidity, etc.; seconly, the abnormal engines with small errors may cause very weak sigals, while other signals would be more stronger, which making it hard to detect and find out those signals. It is not smart to conclude that this device could detect engine failures and assist the car driving with all situations. Instead, the authors need to find out specific situations, troubles and problems. Do not try to resolve all questions and troubles with one simple device. Although the authors claim the good performance of their design, the applications are not well demonstrated to present the unique properties, and they tried to exaggerate the advantages for applications in broad aspects.

Reviewer #2 (Remarks to the Author):

no further comments from this reviewer. Thanks

Response to Comments of Review #1
**NCOMMS-22-39836A: Ultra-sensitive ultra-broadband meta-mechanoreceptor via
mechanical frequency-division multiplexing**

Chong Li, Xinxin Liao, Zhi-Ke Peng, Guang Meng, Qingbo He*

Reviewer's Comment:

In this revision, the authors provide explanations for the questions. But I recommend major revision as there are still some concerns listed below.

Authors' Response:

We thank the reviewer for the positive and important comments on our work that helped us improve the quality of this manuscript. All the comments and suggestions have been carefully considered and the responses are provided as follows in detail. The revisions in the revised manuscript and supplementary information are marked in **BLUE** color.

Reviewer's Comment:

1. The micro-motion stimulation (vibration force/energy required for stimulation for this device) for this materials is not clearly characterized. This property is strongly dependent on the materials' mechanical properties and physical properties. The optimization of materials is not demonstrated?

Authors' Response:

We thank the reviewer for the constructive comments on our work. We agree with the reviewer that the micro-motion stimulation (vibration force/energy required for stimulation for this device) for this material need to be further characterized. In the revised manuscript, we have added the characterization of micro-motion stimulation from three aspects: detection limit, micro-motion direction and frequency range. We use the physical quantity of acceleration to characterize the magnitude of the micro-motion stimulation. The minimum amplitude intensity of micro-motion stimulation that can be detected by MMR reaches 10^{-6} g in the order of magnitude. The micro-motion stimulation direction for MMR is d_{33} direction which is along the z-direction in Cartesian coordinates. The frequency range of micro-motion stimulation can be 0-12 kHz by customizing the design of the MMR. Essentially, the characterization properties rely on the design of zero effective mass that is the key to MMA design. We also added analysis about the effect of the materials' mechanical properties (elastic modulus, shear modulus and density) and physical property (dielectric constant) on the zero effective mass in the revised manuscript. Results demonstrate that A change in the elastic modulus causes a change in stiffness and thus changes the zero-mass frequency. The changes in the shear modulus and density of the helical base have no effect on the zero effective mass. The dielectric constant of the piezoelectric stack has no effect on the mechanical properties of the MMR. We selected the photosensitive resin (DSM IMAGE8000) from Royal DSM to make all the spiral substrates by 3D printing. We selected the commonly used piezoelectric ceramic transducer (PZT) for the fabrication material of the piezoelectric stack. We chose standardized metal copper to make copper pillars. We set the material parameters of photosensitive resin, piezoelectric stack and copper pillar as follows: $E_{\text{res}} = 2.5$ Gpa, $E_{\text{pie}} = 117.4$ Gpa and $E_{\text{cop}} = 110.0$ Gpa; shear modulus $G_{\text{res}} = 1.025$ Gpa, $G_{\text{pie}} = 23.5$ Gpa and $G_{\text{cop}} = 38.5$ Gpa; mass density $\rho_{\text{res}} = 1250$ Kg m⁻³, $\rho_{\text{pie}} = 7500$ Kg m⁻³ and $\rho_{\text{cop}} = 8960$ Kg m⁻³. The dielectric constant of the piezoelectric stack was set to 1433.6. These material properties are determined through systematic study combined with the existing fabrication technology, which is close to the optimal design.

Specifically, we characterize the micro-motion stimulation from three aspects, including **detection limit, micro-motion direction** and **frequency range**, as follows:

First, for the detection limit: as shown in **Figure R1a-c**, the MMR combines unit cells with mechanics-guided piezoelectric resonator design to bypass the limitations of intrinsic piezoelectric materials and achieve high sensitivity. After previous experiments and verification, we use the physical quantity of acceleration to characterize the magnitude of the micro-motion stimulation, and have calculated that MMR has a maximum sensitivity of $36.540 \text{ mv m}^{-1}\text{s}^2$ (358 mv/g). Because the charge output of MMR is the largest at 1540 Hz as shown in **Fig. 2f** of the main text, we applied a single-frequency excitation of 1540Hz with different strengths to the MMR as shown in **Figure R1d**. Then, we estimate the detection limit by extrapolating the data from **Figure R1e**. As shown in **Figure R1d**, we obtained that when the micro-motion stimulation is $3.6 \times 10^{-2} \text{ g}$ and $3.6 \times 10^{-3} \text{ g}$, the corresponding voltage outputs are 12.89 mv and 1.29 mv , respectively. At the same time, a signal with the smallest amplitude of $1 \times 10^{-4} \text{ mv}$ may be detected, corresponding to a detection level of $0.28 \times 10^{-6} \text{ g}$ as shown in **Figure R1e**. Here, the detection estimation is calculated as the ratio of the observed smallest amplitude to the sensitivity of MMR. Thus, the minimum amplitude intensity of micro-motion stimulation that can be detected by MMR can reach 10^{-6} g in the order of magnitude.

Second, for the micro-motion direction: in conventional piezoelectric materials, the electromechanical conversion is described by existing piezoelectric tensors defined under quasi-static conditions. As an example, d_{33} describes the correlation between the stress and the electric displacement in the intrinsic piezoelectric material along the z -direction in Cartesian coordinates. However, the d_{33} defined under quasi-static constraints is not sufficient to describe the piezoelectric property of the unit cell with zero effective mass under dynamic frequency-varying conditions. We define the frequency-dependent effective piezoelectric coefficient $d_{33\text{eff}}(\omega)$ based upon elastodynamics theory to describe the electromechanical conversion of the unit cell in different frequencies as shown in **Supplementary Note 1**:

$$d_{33\text{eff}}(\omega) = A(\omega) d_{33} \frac{1}{m_{\text{eff}}(\omega)} \quad (\text{R1})$$

where $A(\omega)$ represents a frequency-dependent variable related to the dynamic parameter of the unit cell, $m_{\text{eff}}(\omega)$ denotes the effective mass of unit cell and ω denotes the varying frequency. Furthermore, we applied micro-motion excitation to the unit cell in different directions, and confirmed that the unit cell was mono responded in d_{33} direction as shown in **Supplementary Note4**. Thus, the micro-motion stimulation direction for MMR is d_{33} direction which is along the z -direction in Cartesian coordinates.

Third, for the frequency range: in the structural design, MMR is comprised of piezoelectric resonators with distributed zero effective masses featuring a broad range of local resonances. As shown in **Figure R1f**, we have demonstrated that the local resonance frequency could be tuned by varying the structural parameter θ_n . To further demonstrate the bidirectional expansion of the bandwidth, we extended the MMR to 4×4 unit cells based on the original 3×3 unit cells denoted by the white box as shown in **Supplementary Fig. 6a**. We demonstrated that the frequency band Δb_n allowed for a tailored design of center zero-mass frequency in $0\text{-}12 \text{ kHz}$ by individually tuning the parameter θ_n of the unit cell from 8.0 to 17.5 as shown in **Supplementary Note 2** and **Supplementary Fig. 6b**. The zero-mass frequencies corresponding to the unit cells with θ_n of 8.0 and 17.5 are 0.05 kHz and 12 kHz , respectively. Thus, the frequency range of micro-motion stimulation can be $0\text{-}12 \text{ kHz}$ by customizing the design of the MMR.

For all above, we characterize the micro-motion stimulation from three aspects including detection limit, micro-motion direction and frequency range. For the working mechanism, the design of zero effective mass $m_{\text{eff}}(\omega)$ is the core that directly determines the effective piezoelectric coefficient $d_{33\text{eff}}(\omega)$, thus determining the sensitivity and detection limit. The structural design enables the unit

cell to have zero effective mass only in the z direction, which in turn makes the direction of micro-motion stimulation characterized as the d_{33} direction. Moreover, the bandwidth is extendable towards both low-frequency and high-frequency ranges through tuning the zero-mass frequency that directly frequency range of micro-motion stimulation, respectively. Therefore, the characterization properties essentially rely on the design of zero effective mass.

Figure R1 | Characterization of micro-motion stimulation on MMR. **a**, MMR. **b**, unit cell. **c**, Z-direction local resonant response. **d**, Frequency spectrum of MMR at 1540 Hz with a stimulation of $3.6 \times 10^{-2} \text{ g}$ and $3.6 \times 10^{-3} \text{ g}$ in the z -direction. **e**, Detection limit estimation. **f**, The bidirectional expansion of the working frequency band in 0-12 kHz.

As analyzed above, the zero-mass property essentially determines the characterization of micro-motion stimulation. To demonstrate the relationship between the material properties and characterization properties, we further analyze the effect of material properties on the zero-mass frequency ω_0 . To clearly present the details of mechanical and physical properties, we have added a new Supplementary Note 3 to the revised supplementary information as follows:

“We further analyzed the effect of material properties on the zero-mass frequency ω_0 . From **Supplementary Note 2**, combined with the existing mature fabrication technology, the materials of the copper pillar and the piezoelectric stack are fabricated by metal copper and piezoelectric ceramic transducer (PZT) respectively. The mechanical properties of the copper pillars and the piezoelectric stack mainly affect the central mass m of the resonator and have no effect on the stiffness k_n as shown in **Supplementary Equation S12**. Compared with the central mass adjustment, the stiffness adjustment is more efficient in regulating the ω_0 in terms of cost reduction, fabrication simplification and size reduction. For MMR, our design is to change k_n by adjusting the structural parameter θ_n of the spiral base of the unit cell, thus adjusting the ω_0 . Next, we further selected four main material properties for analysis: three mechanical properties are elastic modulus E , shear modulus G and density ρ of the material used in the spiral base. Another physical property is the relative dielectric constant ε of the piezoelectric stack.

We present the results of the material property study in **Supplementary Fig. 2a-d**. Here we change the shear modulus G by adjusting Poisson's ratio μ , where $G = E/2(1+\mu)$ is satisfied. We set the basic parameters of the unit cell as: $E = 2.5 \times 10^9$ Pa, $\mu = 0.41$, $\rho = 1250$ kg/m³ and $\varepsilon = 1433.6$. First, we adjust E from 1×10^9 Pa to 10×10^9 Pa while other parameters remain unchanged, and calculate the zero-mass frequency ω_0 of the unit cell in the finite element analysis. From the numerical results as shown in **Supplementary Fig. 2a**, the ω_0 increases with the increase of E , which is caused by the increase of the k_n due to the increase of E . Then, we set μ from 0.3 to 0.45 as shown in **Supplementary Fig. 2b**, and found that ω_0 remains approximately unchanged with the increase of μ . This is because an increase in μ has no change in stiffness k_n , which also means that only changes in the shear modulus have no effect on the stiffness k_n . Similarly, as shown in **Supplementary Fig. 2c**, we set ρ from 800 kg/m³ to 1700 kg/m³, and also found that ω_0 remains approximately unchanged with the increase of ρ . This is because an increase in ρ only causes an increase in the base mass M and no change in stiffness k_n . Combined with the analysis of **Supplementary Equation S12**, the change of M has little effect on the change of ω_0 . Finally, we set ε from 1000 to 2000 as shown in **Supplementary Fig. 2d**, and found that ω_0 still remains approximately unchanged with the change of ε . This is because the variation of the dielectric constant of the piezoelectric stack has no effect on the mechanical properties of the unit cell with mechanics-guided design.

From the results shown in **Supplementary Fig. 2**, we have systematically studied the effects of E , μ , ρ and ε on the ω_0 . Our results are as follows: ω_0 increases significantly with the increase of E ; ω_0 remains approximately unchanged with the change of μ , ρ and ε . For material's mechanical properties, this means that the change in the elastic modulus causes a change in stiffness and thus changes the zero-mass frequency. The changes in the shear modulus and density of the helical base have no effect on the zero effective mass. Considering the design of zero-mass frequency in the kilohertz frequency band, we choose non-metal as the material of the spiral base, so that the zero-mass design can be completed with a smaller θ_n (from 8.0 to 17.5) to reduce the overall structure size. If the metal is used as the base material, such as iron, the 10 times larger θ_n is required to design a bandwidth of 0-12 kHz. The large θ_n will enlarge the overall structure size of the MMR, and the difficulty of manufacturing the metal helix makes the entire fabrication process very complicated. After comprehensive consideration, we selected the photosensitive resin (DSM IMAGE8000) from Royal DSM to make all the spiral substrates by 3D printing, which can control the fabrication error within 0.1 mm. Since DSM IMAGE8000 has mature fabrication technology, small fabrication error and low fabrication cost, the material selection for the spiral base is close to optimal combined with the existing fabrication technology. For material's physical properties, the dielectric constant of the piezoelectric stack has no effect on the mechanical properties of the MMR. The piezoelectric stack only acts as a transducer in the MMR. The choice of dielectric constant is

not the focus of our work because our work focuses on a mechanics-guided design. Therefore, we selected the most commonly used PZT for the material selection of the piezoelectric stack which has high signal output, stable working performance and mature fabrication technology.

In summary, we have further systematically analyzed the effect of the materials' mechanical properties and physical properties on the zero effective mass. A change in its elastic modulus causes a change in stiffness and thus changes the zero-mass frequency. The changes in the shear modulus and density of the helical base have no effect on the zero effective mass. The dielectric constant of the piezoelectric stack has no effect on the mechanical properties of the MMR. We selected the photosensitive resin (DSM IMAGE8000) to make the spiral bases by 3D printing. We selected the commonly used PZT to fabricate the piezoelectric stack. We chose standardized metal copper to make copper pillars. We set the material parameters of photosensitive resin, piezoelectric stack and copper pillar as follows: $E_{res} = 2.5$ Gpa, $E_{pie} = 117.4$ Gpa and $E_{cop} = 110.0$ Gpa; shear modulus $G_{res} = 1.025$ Gpa, $G_{pie} = 23.5$ Gpa and $G_{cop} = 38.5$ Gpa; mass density $\rho_{res} = 1250$ Kg m⁻³, $\rho_{pie} = 7500$ Kg m⁻³ and $\rho_{cop} = 8960$ Kg m⁻³. The dielectric constant of the piezoelectric stack was set to 1433.6. These material properties are determined through systematic study combined with the existing fabrication technology, which is close to the optimal design.” (Supplementary Note 3)

Additionally, in the revised supplementary information, we have plotted one new **Supplementary Fig. 2** to better present the study on mechanical and physical properties:

Supplementary Fig. 2 | Study on mechanical and physical properties. a, The effect of E on ω_0 . **b,** The effect of μ on ω_0 . **c,** The effect of ρ on ω_0 . **d,** The effect of ϵ on ω_0 .

In addition, the major points discussed above have been included in the revised manuscript addressing this issue as follows:

“We characterized the micro-motion stimulation by detecting micro-motions under different intensities, directions, frequency range and SNRs. For the detection limit, we demonstrated that the lowest detection limit was estimated to reach 10^{-6} g in the order of magnitude which was shown in Supplementary Note 8. For the micro-motion range, we also demonstrated that the direction range of micro-motion was d_{33} direction which was along the z -direction in Cartesian coordinates. The frequency range of micro-motion could be 0-12 kHz by customizing the design of MMR. Moreover, through comparative experiments, we verified that the lowest SNR at which MMR was able to detect micro-motions is -20 dB as shown in Supplementary Note 13, which could be combined with deep learning to have a nearly 100% recognition accuracy.” (Page 15, Line 335)

“..... we set the material parameters of photosensitive resin, piezoelectric stack and copper pillar as follows: $E_{\text{res}} = 2.5$ Gpa, $E_{\text{pie}} = 117.4$ Gpa and $E_{\text{cop}} = 110.0$ Gpa; shear modulus $G_{\text{res}} = 1.025$ Gpa, $G_{\text{pie}} = 23.5$ Gpa and $G_{\text{cop}} = 38.5$ Gpa; mass density $\rho_{\text{res}} = 1250$ Kg m^{-3} , $\rho_{\text{pie}} = 7500$ Kg m^{-3} and $\rho_{\text{cop}} = 8960$ Kg m^{-3} . The piezoelectric tensor d_{33} and dielectric constant of the piezoelectric stack were set to 554 pC N^{-1} and 1433.6, respectively.” (Page 16, Line 353)

“Preparation of MMR began with the fabrication of unit cells. We determined the structural parameters through systematic parameter design combined with the existing fabrication technology. We first fabricated the spiral base of the unit cell with photosensitive resin. In order to ensure the fabrication accuracy and strength of the spiral beam while keeping the size as small as possible, we designed the parameters as $h_3 = 2.5$ mm and $a = 10$ mm. The maximum range of θ_n was 8.0 to 17.5 in a square base of 10 mm by 10 mm. Then we assembled the piezoelectric stack and the copper pillar to the spiral base as shown in Fig. 1d, which was to construct a piezoelectric resonator as a unit cell. For copper pillars and piezoelectric stacks, we designed the parameters as follows in combination with current commercial machining and sintering processes: $h_1 = 3.64$ mm and $h_2 = 5$ mm. The section of the piezoelectric stack was a square with a side length of 1.65 mm, and the section of the copper column was a circle with a radius of 2.2 mm. We fabricated unit cells with different θ_n and combined them into a scalable plane array as the MMR. In the experiments, the constructed MMR consisting of 3×3 unit cells was shown in Supplementary Fig. 7, and the parametric details were presented in Supplementary Table 1. The details of structural parametric design in MMR were shown in Supplementary Note 2. We selected the photosensitive resin (DSM IMAGE8000) to make the spiral bases by 3D printing. We selected the commonly used piezoelectric ceramic transducer (PZT) to fabricate the piezoelectric stack. We chose standardized metal copper to make copper pillars. The details of material property study in MMR were shown in Supplementary Note 3.” (Page 14, Line 315)

2. The device used very small pixel and resolution, however, the motion stimulation would cause the whole device to vibrate and hard to distinguish the motion from various direction and position, if the force is parallel to bottom base. Then the small pixel ad resolution would be meaningless. The authors need to define their limited applications.

Authors' Response:

We thank the reviewer for the constructive comments on our work. MMR behaves as a sensor to collect micro-motion signals, which itself does not have the ability to distinguish the micro-motion from different directions and different positions. When used for distinguishing micro-motions, it is necessary to combine post-processing algorithms, such as deep learning for feature recognition, to analyze the measured signal components and complete signal identification. The application scene of MMR is suitable for single-point and single-axis measurement of micro-motions. Single-point

measurement means that the MMR is positioned at a certain point to measure micro-motions at that point. Single-axis measurement means that the MMR measures micro-motions in one direction, which is along the z -direction in Cartesian coordinates for MMR. Here, MMR plays the role of micro-motion signal acquisition, and its acquisition quality directly determines the micro-motion perception ability. For the focus of our work, the signal acquisition quality is reflected in the sensitivity and frequency bandwidth, and the complexity of the signal is reflected in the complexity of frequency components and frequency bands. For the contribution of our work, MMR addresses the trade-off between sensitivity and bandwidth in conventional mechanics-guided micro-motion sensors.

In addition, if the unit cell of MMR is regarded as a structural pixel, the plane size of each pixel is 10 mm by 10 mm, and the working frequency band of the n th pixel is Δb_n . By individually tuning the θ_n of the unit cell, the Δb_n allowed for a tailored design of zero-mass frequency value. When MMR detects micro-motions, the sampling frequency f_s and the number N of sampling points can be set as required. The f_s in our experiments is 20 kHz, and the N is more than 4×10^5 . We calculated that the frequency resolution (f_s/N) of the Fourier transform is 0.05 Hz, which is a high frequency resolution for signal processing. We applied swept micro-motion excitation to MMR in d_{33} direction, and observed that MMR featured a broad range of local resonances around the zero-mass frequencies. As shown in **Figure R1c**, the vibration during local resonance is mainly concentrated on the central mass of the unit cell rather than the whole device. We agree with the reviewer that the MMR will not be able to sense micro-motion stimulation when it is parallel to the bottom base. This is because MMR is mono responded in d_{33} direction. To clearly explain the details of the mono response of MMR in d_{33} direction, we have added a new **Supplementary Note 4** to the revised supplementary information as follows:

“Since the practical micro-motions are multi-directional, we further give the explanation to approve MMR is mono responded in d_{33} direction as shown in **Supplementary Fig. 3**. Here, d_{33} describes the correlation between the stress and the electric displacement in the intrinsic piezoelectric material along the z -direction in Cartesian coordinates. As shown in **Supplementary Fig. 3a**, the piezoelectric stack is between the spiral base and the copper pillar, and converts the micro-motion excitation from the spiral base into a voltage signal. The piezoelectric stack adopts the z -direction stacking manufacturing process, and we have tested that only the z -direction charge output is generated. Since the piezoelectric stack is installed on the helical base, applying weak vibration excitations in different directions to the helical base can only cause the piezoelectric stack to produce voltage outputs in the z -direction. Here, the micro-motions in different directions can cause the stress or shear force in different directions in the spiral base. As shown in **Supplementary Fig. 3b**, we list all 6 possible micro-motion directions applied to the spiral base, including stress direction along x axis (denoted as 1), stress direction along y axis (denoted as 2), stress direction along z axis (denoted as 3), shear force direction rotating around x axis (denoted as 4), shear force direction rotating around y axis (denoted as 5) and shear force direction rotating around z axis (denoted as 6). Since the n th unit cell only has voltage output in the z -direction (denoted as 3), only six directions between d_{11}^n and d_{66}^n are valid for MMR, and the six directions are d_{31}^n , d_{32}^n , d_{33}^n , d_{34}^n , d_{35}^n and d_{36}^n , respectively. We present the numerical simulations of six different directions with finite element analysis method as shown in **Supplementary Figs. 3c-h**, respectively. We apply swept-frequency micro-motion excitations of equal amplitude and different directions to the spiral base, and then calculate the voltage output of the unit cell in the z -direction.

As shown in **Supplementary Fig. 4**, we present voltage output details for different directions and normalize the output amplitude of the voltage. Taking the d_{33}^n direction as the normalized benchmark, the maximum output amplitude value of d_{33}^n is 1, while the maximum output amplitude values in the d_{31}^n , d_{32}^n , d_{34}^n , d_{35}^n and d_{36}^n directions are 2.2e-3, 2.05e-3, 1.4e-3, 1.8e-3 and 8e-3,

respectively. From the results, the voltage output in the other five directions is far smaller than the voltage output in the direction of d_{33} , which can be approximated that MMR is mono responded in d_{33} direction.” (Supplementary Note 4)

In addition, the major points discussed above have been included in the revised manuscript addressing this issue as follows:

“When the unit cell was excited by external micro-motions (finite F) at the zero-mass frequency, the dynamic response was enhanced by the zero effective mass ($m_{\text{eff}}(\omega) \rightarrow 0$) with a near-infinite acceleration amplitude ($F/m_{\text{eff}}(\omega) \rightarrow \infty$). We applied micro-motion excitation to the unit cell in different directions, and confirmed that the unit cell was mono responded in d_{33} direction (Supplementary Note 3). We calculated the $m_{\text{eff}}(\omega)$ and $d_{33\text{eff}}(\omega)$ of the unit cell with the numerical method, which showed excellent agreement with the analytical results (see the red area in Fig. 2b). Due to the inevitable damping in experiments, the maximum $d_{33\text{eff}}(\omega)$ reached an appreciable value of 24930 pC N⁻¹ instead of a theoretical near-infinite value (Fig. 2d), which was much higher than that of the state-of-art piezoelectric materials⁴⁹⁻⁵⁷. Therefore, configuring zero effective mass design in the unit cell results in a near-infinite effective piezoelectric coefficient, allowing access to micro-motion sensing with an ultrahigh sensitivity around the zero-mass frequency.” (Page 7, Line 146)

In summary, we have added data and explanations to demonstrate that MMR is mono responded in d_{33} direction which is along the z -direction in Cartesian coordinates. The application scene of MMR is suitable for single-point and single-axis measurement of micro-motions. For our current designed MMR, there are still many unexplored problems when being applied in practical applications. This is expected to be solved in future research. We have also added explanation of limited applications to the **Main text** addressing this issue as follows:

“Experimental results demonstrate a proof-of-concept application of MMR for micro-motion sensing. Nevertheless, our current study also has certain limitations. The application scene of MMR is suitable for single-point and single-axis measurement of micro-motions. The detection direction of MMR is limited to the d_{33} direction. The MMR can also be used for detection of complex micro-motion that has a component projected in the d_{33} direction. Further research on MMR can expand the spatial distribution of unit cells to three-dimensional configurations, which can be optimized for improved sensing performance in multiple directions⁶³. The Lego-like heterogeneously constructed metamaterials provide a high degree of freedom for tailoring the sensitivity and bandwidth of MMR, which can be combined with stimuli-responsive metamaterials for adaptive sensing in different frequency ranges^{64,65}. By using advanced micro and flexible manufacturing technology^{66,67}, the width and height of MMR can be made in subwavelength, which is hoped to be integrated into various flexible micro-electro-mechanical systems. By matching the integrated central processing unit (CPU), MMR can be packaged as an all-in-one system with sensing, storage and computing functions (Supplementary Note 19), which can enable the entire sensing system to be simpler and more portable⁶⁸⁻⁷⁰. MMR can also provide high versatility for complex micro-motion scenes by incorporating artificial intelligence (AI) techniques, which can be applied to sensing and computing in edge AI tasks⁷¹.” (Page 13, Line 293)

In the revised supplementary information, we have plotted one new **Supplementary Fig. 3** to better present the response simulation of unit cell under excitation in different directions as follows:

Supplementary Fig. 3 | Response simulation of unit cell under excitation in different directions. **a**, The unit cell model. **b**, Six possible directions of micro-motion excitation on the spiral base. **c**, Simulation in six different directions corresponding to (c) “ d_{31}^n ”, (d) “ d_{32}^n ”, (e) “ d_{33}^n ”, (f) “ d_{34}^n ”, (g) “ d_{35}^n ” and (h) “ d_{36}^n ”.

In the revised supplementary information, we have also plotted one new **Supplementary Fig. 4** to better present the response details of unit cell in different directions as follows:

Supplementary Fig. 4 | Response details of unit cell in different directions. The voltage output amplitude of the unit cell in six directions corresponds to (a) “ d_{31} ”, (b) “ d_{32} ”, (c) “ d_{33} ”, (d) “ d_{34} ”, (e) “ d_{35} ” and (f) “ d_{36} ” under equal-amplitude swept-frequency micro-motion excitation.

3. There is no enough introduction of the conventional piezoelectric mechanoreceptor. Is this a commercial product? If it is, then they need to provide the details of the item, such as model/product number and manufacture name. If this is made by the authors, they need to introduce the details of the manufacturing processes, and compare with other products if possible. If the information of the conventional one is missing, it is hard to say their performance are superior to all other mechanoreceptors.

Authors' Response:

We thank the reviewer for the careful and constructive comments on our work. The conventional piezoelectric mechanoreceptor is a commercial product with non-resonant design that is suitable for single-point and single-axis vibration measurement. The product number is B&K4371 and the manufacture name is Hottinger Brüel & Kjær (HBK) company. As shown in **Fig. 1g** of the main text, the working frequency range of conventional piezoelectric mechanoreceptor is in the non-resonant range far from the resonance frequency. The broadband micro-motion excitations are not enhanced and thus difficult to be detected. In our experiments, we use the conventional non-resonant piezoelectric mechanoreceptor as a comparison to verify the working performance of MMR. Due to the non-resonant design, the conventional piezoelectric mechanoreceptor shown in **Fig. 1g** of the main text has an amplification factor of 1 in the working frequency band. This type of non-resonant piezoelectric sensor has been commercialized in HBK. Here, we chose the commonly used B&K4371 from HBK as an example in the experiment.

In the revised manuscript, we have added the corresponding product and manufacture names of the instruments and sensors used in all experiments, which is provided in the main text as follows:

“For the experimental $d_{33\text{eff}}^n(\omega)$ as shown in Fig. 2f, we first constructed a swept harmonic signal through a signal generator (SDG2122X, SIGLENT). Then, the constructed signal was amplified by a power amplifier (YE5872A, Sinocera Piezotronics INC) to drive the exciter (B&K4808, HBK) for generating micro-motion excitation $a(t)$. We fixed the MMR to the exciter, and the micro-motion excitation activated electric displacement of the unit cell in the z -direction. We also fixed the conventional piezoelectric mechanoreceptor (B&K4371, HBK) on the exciter as a comparison. We extracted the piezoelectric voltage $v_n(t)$ of each measurement channel through the data acquisition equipment (DH8300N, Donghua Jiaozhun)” (Page 16, Line 365)

4. The detection limit and range of the motion is not demonstrated. The mechanical motions can be different with different styles of vibration and energy levels. The authors need to characterize all the details, and compare if possible.

Authors' Response:

We thank the reviewer for the constructive comments on our work. We agree with the reviewer that we need to further characterize all details including the detection limit and range of the motion. In the revised manuscript, we have compared and analyzed the output of MMR under different intensities of micro-motion stimulation. For the detection limit, we demonstrate that the lowest detection limit is estimated to reach 0.28×10^{-6} g. For the micro-motion range, we also demonstrate that the direction range of micro-motion is d_{33} direction which is along the z -direction in Cartesian coordinates. The frequency range of micro-motion can be 0-12 kHz by customizing the design of the MMR. Moreover, through comparative experiments, we verified that the lowest signal-to-noise ratio at which MMR is able to detect micro-motions is -20 dB, which can be combined with deep learning to have a nearly 100% recognition accuracy.

First, to clearly present the details of detection limit, we have added a new **Supplementary Note 8** to the revised supplementary information as follows:

“In our experiments, we use the physical quantity of acceleration to characterize the magnitude of the micro-motion stimulation, and have calculated that MMR has a maximum sensitivity of $36.540 \text{ mv m}^{-1}\text{s}^2$ in the main text. Because the charge output of MMR is the largest at 1540 Hz as shown in **Fig. 2f** of the main text, we applied a single-frequency excitation of 1540 Hz with different strengths to MMR as shown in **Supplementary Fig. 10a**. Then, we estimate the detection limit by extrapolating the data from **Supplementary Fig. 10b**. As shown in **Supplementary Fig. 10a**, we obtained that when the micro-motion stimulation is $3.6 \times 10^{-2} \text{ g}$ and $3.6 \times 10^{-3} \text{ g}$, the corresponding voltage outputs are 12.89 mv and 1.29 mv, respectively. At the same time, a signal with the smallest amplitude of $1 \times 10^{-4} \text{ mv}$ may be detected, corresponding to a detection level of $0.28 \times 10^{-6} \text{ g}$ as shown in **Supplementary Fig. 10b**. For comparison, we further decreased the intensity of micro-motion stimulation, and then observed the voltage output of MMR. As shown in **Supplementary Fig. 10c**, we obtained that when the micro-motion stimulation is $6.0 \times 10^{-4} \text{ g}$ and $6.0 \times 10^{-5} \text{ g}$, the corresponding voltage outputs are 0.21 mv and 0.02 mv, respectively. At the same time, a signal with the smallest amplitude of $1 \times 10^{-4} \text{ mv}$ can also be detected, corresponding to a detection level of $0.28 \times 10^{-6} \text{ g}$ as shown in **Supplementary Fig. 10d**. By comparing **Supplementary Figs. 10b** and **10d**, we found that the signal with minimum amplitude of $1 \times 10^{-4} \text{ mv}$ can both be observed under different intensities of micro-motion stimulation, which verifies that the lowest detection limit can reach $0.28 \times 10^{-6} \text{ g}$. Thus, the lowest detection limit is estimated to reach 10^{-6} g in the order of magnitude.” (Supplementary Note 8)

In the revised supplementary information, we also have plotted one new **Supplementary Fig. 10** to better present the detection limit study as follows:

Supplementary Fig. 10 | Detection limit study. **a**, Frequency spectrum of MMR at 1540 Hz with a stimulation of $3.6 \times 10^{-2} \text{ g}$ and $3.6 \times 10^{-3} \text{ g}$ in the z-direction and **(b)** detection limit estimation. **c**, Frequency spectrum of MMR at 1540 Hz with a stimulation of $6.0 \times 10^{-4} \text{ g}$ and $6.0 \times 10^{-5} \text{ g}$ in the z-direction and **(d)** detection limit estimation.

Second, for the micro-motion range: as shown in Supplementary Note 4, we applied micro-motion excitation to the unit cell in different directions, and confirmed that the unit cell was mono responded in d_{33} direction. Thus, the micro-motion direction range for MMR is d_{33} direction which is along the z -direction in Cartesian coordinates. From the detection limit study as shown in Supplementary Fig. 10, a signal with the smallest amplitude of 1×10^{-4} mv can be detected, corresponding to a detection level of 0.28×10^{-6} g. The maximum vibration measurement for MMR depends on the vibration displacement limit of the unit cell. Due to structural design and strength limitations, assuming that the displacement response of the unit cell does not exceed 0.1mm, combined with the magnification factor shown in Fig. 3c of the main text, we estimate that the maximum detection limit is 21.26 g which is calculated by second derivative of displacement. Similarly, we also calculated the values of maximum detection limit were 5.31 g, 10.47 g, 12.48 g, 19.16 g, 50.85 g, 84.97 g, 132.56 g and 233.50 g under other frequencies which were corresponding to 437 Hz, 684 Hz, 964 Hz, 1250 Hz, 1900 Hz, 2130 Hz, 2480 Hz and 2880 Hz, respectively. Since MMR is suitable for micro-motion application scenarios, the lowest detection limit is the focus and is estimated to reach about 10^{-6} g in the order of magnitude. The maximum detection amplitude is a frequency-dependent variable that can be calculated. As shown in Figure R1f and Supplementary Fig. 5, we demonstrated that the frequency band Δb_n allowed for a tailored design of zero-mass frequency in 0-12 kHz by individually tuning the parameter θ_n from 8.0 to 17.5. thus, the frequency range of micro-motion stimulation can be 0-12 kHz by customizing the design of the MMR. Therefore, for micro-motion stimulation, the direction range is d_{33} direction, the lowest detection limit is about 10^{-6} g in the order of magnitude, and the frequency range can be 0-12 kHz.

Third, for the signal-to-noise ratio (SNR): The mechanical micro-motions can be different with different SNRs, we further validated the lowest SNR at which MMR was able to detect micro-motions. To clearly present the comparison details of recognition accuracy under different SNRs, we have added a new Supplementary Note 13 to the revised supplementary information as follows:

“Considering that the mechanical micro-motions can be different with different styles of vibration and energy levels, we evaluated the lowest signal-to-noise ratio (SNR) at which MMR was able to detect micro-motion signals. As shown in Supplementary Note 11, we constructed an intelligent sensing system by combining MMR with a deep learning technique. The micro-motion signal was measured by MMR and then was transported to a deep neural network to complete the signal recognition. Here, we validated the performance of MMR by comparing the recognition rates under different SNRs. From Fig. 4b of the main text, when the SNR was -20 dB, we have demonstrated that the recognition accuracies of patterns with and without MMR were about 99.5% and 34.0%, respectively. We constructed four different signals according to the signal components shown in Supplementary Table 5, and completed verification experiments at -25 dB, -30 dB, -35 dB and -40 dB, respectively. The comparison of recognition accuracy under different SNRs with and without MMR is shown in Supplementary Fig. 18. The details of recognition accuracy under different SNRs with and without MMR are shown in Supplementary Figs. 19 and 20, respectively. From the results shown in Supplementary Fig. 18, we found that the recognition accuracy with MMR decreases as the SNR decreases. Specifically, when the SNRs are -20 dB and -45 dB, the recognition accuracies with MMR are 99.5 % and 58.0%, respectively. This is because the signal components that can be measured by MMR decrease as the SNR decreases. From the results without MMR shown in Supplementary Fig. 18, we found that the recognition accuracy without MMR fluctuates below 40% as the SNR decreases. This is because the conventional piezoelectric mechanoreceptor cannot measure clear signal components since the signal with SNR below -20 dB cannot reach the detection limit. Overall, when the SNRs are -20 dB and -30 dB, the recognition accuracies with MMR are 99.5 % and 83.5%, respectively. To ensure a near 100% accuracy rate, we conclude that the lowest SNR at which MMR is able to detect micro-motions is -20 dB.”

In the revised supplementary information, we also have plotted one new **Supplementary Fig. 18** to better present the comparison of recognition accuracy under different SNRs as follows:

Supplementary Fig. 18 | Comparison of recognition accuracy under different SNRs with and without MMR.

In the revised supplementary information, we also have plotted one new **Supplementary Fig. 19** to better present the details of recognition accuracy under different SNRs with MMR as follows:

Supplementary Fig. 19 | Details of recognition accuracy under different SNRs with MMR. The recognition accuracy corresponding to (a) “-25 dB”, (b) “-30 dB”, (c) “-35 dB” and (d) “-40 dB” with MMR.

In the revised supplementary information, we also have plotted one new **Supplementary Fig. 20** to better present the details of recognition accuracy under different SNRs without MMR as follows:

Supplementary Fig. 20 | Details of recognition accuracy under different SNRs without MMR. The recognition accuracy corresponding to (a) “-25 dB”, (b) “-30 dB”, (c) “-35 dB” and (d) “-40 dB” without MMR.

In addition, the major points discussed above have been included in the revised manuscript addressing this issue as follows:

“We characterized the micro-motion stimulation by detecting micro-motions under different intensities, directions, frequency range and SNRs. For the detection limit, we demonstrated that the lowest detection limit was estimated to reach 10^{-6} g in the order of magnitude which was shown in Supplementary Note 8. For the micro-motion range, we also demonstrated that the direction range of micro-motion was d_{33} direction which was along the z-direction in Cartesian coordinates. The frequency range of micro-motion could be 0-12 kHz by customizing the design of MMR. Moreover, through comparative experiments, we verified that the lowest SNR at which MMR was able to detect micro-motions is -20 dB as shown in Supplementary Note 13, which could be combined with deep learning to have a nearly 100% recognition accuracy.” (Page 15, Line 335)

5. The vibration and motions can be complex, while this device can convert one simple motion into more complex vibrations, such as rotation, translation. Then this device would not be able to detect the movement states, which making the device less functional.

Authors' Response:

We thank the reviewer for the constructive comments on our work. We agree with the reviewer that the vibration and micro-motions during MMR detection can be complex. In this work, we developed MMR to address the trade-off between sensitivity and bandwidth in conventional mechanics-guided micro-motion sensors. MMR plays the role of micro-motion signal acquisition, and its acquisition quality directly determines the micro-motion perception ability. For the focus of our work, the complexity of the signal is reflected in the complexity of frequency components. For the working mechanism of MMR, the micro-motion information is reflected in the frequency spectrum, and MMR relies on the collected frequency components to detect micro-motions. For the demonstrated bearing failure detection shown in **Figure R2**, the frequency feature components corresponding to the normal state and three typical kinds of bearing faults are complex and different. For the demonstrated engine failure detection as shown in **Figure R3**, the frequency feature components corresponding to the normal state and five typical kinds of engine failures are also complex and different. From the measured results, the complexity of typical bearing faults and engine faults is reflected in the complexity of the frequency feature components of the signal, rather than the complexity of rotation or translation. Moreover, we have added a demonstration that MMR can detect three movement states of the rotor test bench during the start-up process. We explain that the realization of micro-motion sensing by MMR needs to meet the following two conditions: one is that the micro-motion has a component projected in the corresponding z direction (d_{33} direction of MMR); the other is that the intensity reaches the detection limit of MMR and the frequency is within the detection frequency range for the micro-motion component.

Figure R2 | Details of signal measurements in structural health monitoring. The measured vibration signals from the bear, including one normal signal and three fault signals (Fault #1-#3). The signal measurements corresponding to (a) “Normal”, (b) “Fault #1”, (c) “Fault #2” and (d) “Fault #3” with and without MMR.

Figure R3 | Details of signal measurements in smart-driving assistance. **a**, Schematic of smart-driving assistance by equipping MMR into the in-vehicle system. The measured vibration signals from the engine, including one normal signal and five fault signals (Fault #1-#5). The signal measurements corresponding to **(b)** “Normal”, **(c)** “Fault #1”, **(d)** “Fault #2”, **(e)** “Fault #3”, **(f)** “Fault #4” and **(g)** “Fault #5” with and without MMR.

Moreover, the vibration and micro-motions during MMR detection can also be complex in directionality. As an example, we demonstrate the use of MMR to detect the movement state of a rotor test bench during the start-up process as shown in **Supplementary Fig. 28**. During the operation of the rotor test bench, the main movement is the rotational movement which generally causes the vibration in the z direction. We mounted the MMR above the bearing block on the right end. Then we start the test bench from 0 RPM to 4000 RPM, and keep it running stably at 4000 RPM. Experimental results demonstrate that even though the vibration source during the start-up process of the rotor test bench is complexed and only z -direction vibration components are detected, the MMR can still successfully identify the three movement states: stationary state; speed up state and steady state. To clearly present the details of running status detection of rotor test bench, we have added a new **Supplementary Note 17** to the revised supplementary information as follows:

“In practical applications, the vibration and micro-motions during MMR detection can be complex. As shown in **Supplementary Figs. 28a** and **28b**, we mainly divide the direction of micro-motion into two types: one is z -direction stimulation, and the other is non- z -direction stimulation. For the z -direction stimulation shown in **Supplementary Fig. 28a**, the direction of micro-motion stimulation is consistent with the d_{33} detection direction of MMR, which is the most beneficial detection condition to maximize the transmission of micro-motion into MMR. For the non- z -direction stimulation shown in **Supplementary Fig. 28b**, the direction of the micro-motion stimulation has a certain angle with the z -direction. In this case, we can decompose the micro-motion S into the z -direction and the parallel direction to obtain S_z and S_o , respectively. Since MMR is mono responded in z -direction, the S_z component can be enhanced with the local resonance as shown in **Supplementary Fig. 28c**, while S_o component cannot be detected. In practical scenes, the micro-motion feature components may be different in different directions. If the S_z contains sufficient signal feature components, we can still use MMR to detect the micro-motion signals.

As shown in **Supplementary Fig. 28d**, we demonstrate the use of MMR to detect the movement state of a rotor test bench during the start-up process. The architecture of the whole rotor test bench is shown in **Supplementary Fig. 25**. We mounted the MMR above the bearing block on the right end. Then we started the test bench from 0 RPM to 4000 RPM, and kept it running stably at 4000 RPM. During start-up, the vibration generated by the rotor was time-varying and multi-directional, which caused the z -direction vibration to be transmitted to the MMR through the bearing block. For MMR, the vibration source was extremely complex, and only the corresponding vibration component in the z direction can be detected. The time-domain and time-frequency presentation (unified) of measured signals during the start-up process were shown in **Supplementary Figs. 28e** and **28f**, from which we found that MMR detected three operating movement states: The first state was a stationary state from 0 to 5 s, during which the rotor test rig was ready to start and there was no signal component; the second state was a speed up state from 5 to 35 s, during which the rotor test bench accelerated from 0 to 4000 RPM and the frequency of signal component moved to high frequency gradually; the third state was the steady state after 35 s, during which the rotor test bench ran stably at 4000 RPM and the signal component was in a stable state.

In summary, we provide a demonstration that MMR could detect three movement states of the rotor test bench during the start-up process. In practical applications, the vibration and micro-motions during MMR detection, especially in directionality, can be complex. We demonstrate that the realization of micro-motion sensing by MMR needs to meet the following two conditions: one is that the micro-motion has a micro-motion signal component in the corresponding z direction (d_{33} direction of MMR); the other is that the micro-motion signal component reaches the detection limit of MMR and is within the frequency range.” (Supplementary Note 17)

In the revised supplementary information, we also have plotted one new **Supplementary Fig. 28** to better present the details of running status detection of rotor test bench as follows:

Supplementary Fig. 28 | Running status detection of rotor test bench. **a**, z-direction stimulation. **b**, non-z-direction stimulation. **c**, local resonance of unit cell. **d**, rotor test bench. The **(e)** time-domain and **(f)** time-frequency presentation of measured signals during the start-up process of the test bench.

6. For the detection of engine failure, the authors need to define the specific situations of failure detection. For sure, it would be very difficult to detect all engine troubles with this device. Firstly, the normal engines sometimes still can generate abnormal signals under bad road conditions, and they would have different signals under different days caused by weather, humidity, etc.; secondly, the abnormal engines with small errors may cause very weak signals, while other signals would be more stronger, which making it hard to detect and find out those signals. It is not smart to conclude that this device could detect engine failures and assist the car driving with all situations. Instead, the authors need to find out specific situations, troubles and problems. Do not try to resolve all questions and troubles with one simple device. Although the authors claim the good performance of their design, the applications are not well demonstrated to present the unique properties, and they tried to exaggerate the advantages for applications in broad aspects.

Authors' Response:

We thank the reviewer for the constructive comments on our work. We agree with the reviewer that we need to define the specific situations for the detection of engine failure. In this application scenario, we conducted experimental demonstrations with six engine vibration signals, including one normal signal and five faulty signals. The five faulty signals come from specific situations including: **the balance shaft on the intake side (Fault #1), the abnormal noise during acceleration (Fault #2), the camshaft on the intake side (Fault #3), the abnormal noise during the low-temperature start (Fault #4), and the four-position three-way valve on the intake side (Fault #5)**. We demonstrate that the typical fault signal can be extracted with MMR, and is processed by the deep neural network to complete the identification of early faults. The working conditions of the signals in the training set and the testing set are consistent, and the quality of signal acquisition is the core which directly determines the recognition accuracy.

Furthermore, for the working mechanism, MMR behaves as a sensor to collect micro-motion signals, which itself does not have the ability to distinguish the micro-motion from different scenarios. When used for distinguishing micro-motions, it is necessary to combine post-processing algorithms, such as deep learning, to analyze the measured signal components and complete signal identification. For the contribution of our work, MMR addresses the trade-off between sensitivity and bandwidth in conventional mechanics-guided micro-motion sensors. MMR plays the role of micro-motion signal acquisition, and its acquisition quality directly determines the micro-motion perception ability. In the demonstrated scenario, we verify that the MMR exhibits micro-motion sensing capabilities that cannot be achieved by conventional piezoelectric mechanoreceptors. When the SNR reaches -20 dB, the MMR can still measure the majority of the signal components, while the conventional piezoelectric mechanoreceptor cannot measure clear signal components since the signal with SNR of -20 dB cannot reach the detection limit. And for MMR, the bandwidth with ultrahigh sensitivity is extendable towards both low-frequency and high-frequency ranges in 0-12 kHz through tuning the local resonance of each individual sensing cell. Thus, the ultra-sensitive ultra-broadband micro-motion sensing property of MMR is unique compared to the conventional piezoelectric mechanoreceptors.

Moreover, we agree with the reviewer that MMR could not detect engine failures and assist the car driving with all situations. To deal with other failure situations, more signal sample libraries need to be established in these situations, and then used for training and identifying the corresponding signal types via deep learning. More failure detection scenarios can also be realized by introducing more advanced deep learning techniques, such as transfer learning, etc. For specific situations, customized design of MMR is also required to match the corresponding detection limit, micro-motion direction, sensitivity and bandwidth, etc. We give the explanation that the micro-motion sensing technology itself has great potential and is urgently needed in broad application scenarios,

and MMR can address ultra-sensitive ultra-broadband micro-motion sensing problems that conventional piezoelectric mechanoreceptors cannot deal with.

In summary, the main contribution of our work focuses on the design principle of the mechanoreceptor. For the design of artificial mechanoreceptors capable of micro-motion sensing, there's a trade-off between the sensitivity and bandwidth of mechanics-guided designs due to the nature of resonance effect. To address this challenge, we have reported a bioinspired MMR for ultra-sensitive ultra-broadband micro-motion sensing via mechanical frequency-division multiplexing in the revised manuscript. MMR addresses the trade-off between sensitivity and bandwidth in conventional mechanics-guided micro-motion sensors. This is achieved by tessellating piezoelectric resonators with distributed zero effective masses into a metamaterial architecture that performs as MMR for micro-motion sensing. MMR exhibits frequency-dependent effective piezoelectric coefficients that break the limits of intrinsic piezoelectric materials in which the electromechanical conversion is described by existing piezoelectric tensors under quasi-static stations. Experiments confirmed that the maximum sensitivity of MMR was improved by two orders of magnitude compared with the conventional piezoelectric mechanoreceptor, and its bandwidth with ultrahigh sensitivity was extendable towards both low-frequency and high-frequency ranges in 0-12 kHz. This manuscript demonstrates three proof-of-concept applications based on MMR, including spatio-temporal sensing, remote-vibration monitoring and smart-driving assistance, and also explores structural health monitoring in industrial scenarios. Besides the applications demonstrated in this manuscript, our work revealed that MMR may be a promising candidate for ultra-sensitive ultra-broadband micro-motion sensing in mechanics-guided sensors, providing a new perspective on simpler mechanics-guided designs of high-performance sensors for various other physical information. For our current designed MMR, there are still many problems that need to be further explored when being applied in practical applications. This is expected to be solved in future research. We have also added corresponding explanations and exploration possibilities to the **Main text** addressing this issue as follows:

“..... Here, the early fault signal can be extracted with MMR in time, and is processed by the deep neural network to complete the identification of early faults, achieving autonomous monitoring and diagnosis. The vehicle executes the decisions based on the identification results, such as fault warning and safety inspection. We conducted experiments with six engine vibration signals, including one normal signal and five typical faulty signals from specific situations (**Fig. 4f**). The working conditions of the signals in the training set and the testing set are consistent, and the quality of signal acquisition is the core which directly determines the recognition accuracy. The recognition accuracy with MMR was 95.3%, which was more than 5 times higher than the accuracy of 14.3% without MMR (Supplementary Note 15). Furthermore, we illustrated that MMR enabled structural health monitoring in the helicopter transmission system by detecting early rotor failures, which was unachievable in the case without MMR (Supplementary Note 16). Therefore, the demonstrated results revealed that MMR was a promising candidate for ultra-sensitive ultra-broadband sensing in the Internet of Vehicles system or other industrial scenarios.” (Page 12, Line 262)

“Experimental results demonstrate a proof-of-concept application of MMR for micro-motion sensing. Nevertheless, our current study also has certain limitations. The application scene of MMR is suitable for single-point and single-axis measurement of micro-motions. The detection direction of MMR is limited to the d_{33} direction. The MMR can also be used for detection of complex micro-motion that has a component projected in the d_{33} direction. Further research on MMR can expand the spatial distribution of unit cells to three-dimensional configurations, which can be optimized for improved sensing performance in multiple directions⁶³. The Lego-like heterogeneously

constructed metamaterials provide a high degree of freedom for tailoring the sensitivity and bandwidth of MMR, which can be combined with stimuli-responsive metamaterials for adaptive sensing in different frequency ranges^{64,65}. By using advanced micro and flexible manufacturing technology^{66,67}, the width and height of MMR can be made in subwavelength, which is hoped to be integrated into various flexible micro-electro-mechanical systems. **By matching the integrated central processing unit (CPU), MMR can be packaged as an all-in-one system with sensing, storage and computing functions (Supplementary Note 19), which can enable the entire sensing system to be simpler and more portable⁶⁸⁻⁷⁰. MMR can also provide high versatility for complex micro-motion scenes by incorporating artificial intelligence (AI) techniques, which can be applied to sensing and computing in edge AI tasks⁷¹.”** (Page 13, Line 293)

We would like to thank the reviewer for the careful and constructive review on our work again. In the revised manuscript and supplementary information, we have carefully considered the reviewers’ suggestions and comments and provided corresponding revisions, hoping that these revisions are satisfactory.

Response to Comments of Review #2
**NCOMMS-22-39836A: Ultra-sensitive ultra-broadband meta-mechanoreceptor via
mechanical frequency-division multiplexing**
Chong Li, Xinxin Liao, Zhi-Ke Peng, Guang Meng, Qingbo He*

Reviewer's Comment:

no further comments from this reviewer. Thanks.

Authors' Response:

We thank the reviewer for the positive and important comments on our work that helped us improve the quality of this manuscript.

Reviewer #2 (Remarks to the Author):

The revision has provided more details on the simulation and results. This reviewer believes that it could clarify the design and usage limitations. However, the most crucial aspect is the theoretical design and numerical simulation being applicable in practice. It is desired to observe if the testing data can offer greater accuracy.

Response to Comments of Review #1
**NCOMMS-22-39836B: Ultra-sensitive ultra-broadband meta-mechanoreceptor via
mechanical frequency-division multiplexing**

Chong Li, Xinxin Liao, Zhi-Ke Peng, Guang Meng, Qingbo He*

Reviewer's Comment:

The revision has provided more details on the simulation and results. This reviewer believes that it could clarify the design and usage limitations. However, the most crucial aspect is the theoretical design and numerical simulation being applicable in practice. It is desired to observe if the testing data can offer greater accuracy.

Authors' Response:

We thank the reviewer for the positive and important comments on our work that helped us improve the quality of this manuscript. All the comments and suggestions have been carefully considered and the responses are provided as follows in detail. The revisions in the revised manuscript and supplementary information are marked in **BLUE** color.

We agree with the reviewer that the theoretical design and numerical simulation being applicable in practice is the most crucial aspect. For MMR, the core design is that configuring frequency-dependent zero effective mass $M_{\text{eff}}(\omega)$ in the unit cell results in a near-infinite effective piezoelectric coefficient $d_{33\text{eff}}^n(\omega)$, allowing access to micro-motion sensing with a high sensitivity around the zero-mass frequency. In the revised manuscript, we calculated the $M_{\text{eff}}(\omega)$ and $d_{33\text{eff}}^n(\omega)$ by performing numerical finite element simulation, which showed excellent agreement with the theoretical design with analytical derivation. We calculated the practical $d_{33\text{eff}}^n(\omega)$ through experimental verification, which also showed excellent agreement with the theoretical design and numerical simulation. Furthermore, we demonstrated that MMR shows strong stability and reliability under long-time measurement of 100 s. We also gave specific experimental data in the demonstrated remote-vibration monitoring, smart-driving assistance and structural health monitoring, which confirmed that MMR showed strong stability and reliability in the measurement of amplitude stability with a zero drift of less than 0.01. Moreover, we demonstrated that the reconstruction accuracy of measured data can be further improved by repeated measurements. Compared with the one-time measurement, the measurement error of five repeated measurements with computational sensing is reduced from 3% to 1% in the demonstrated case. Besides, combining MMR with a deep learning technique, we demonstrated the recognition accuracy of testing data under -25dB can be increased from 92.5% to 99.5% by increasing the number of training samples from 200 to 400.

First, for the theoretical design, numerical simulation and experimental verification: as shown in **Figure R1**, we derived the effective piezoelectric coefficient $d_{33\text{eff}}^n(\omega)$ of the n th unit cell in MMR as follows:

$$d_{33\text{eff}}^n(\omega) = A(\omega) d_{33} \frac{1}{M_{\text{eff}}(\omega)} \quad (\text{R1})$$

where $M_{\text{eff}}(\omega)$ is the frequency-dependent effective mass of MMR and generates distributed zero values in different frequencies as shown in **Figure R1a**. The MMR exhibits near-infinite $d_{33\text{eff}}^n(\omega)$ at these distributed zero-mass frequencies. With the finite element analysis, we applied swept micro-motion excitation to MMR, and observed that MMR featured a broad range of local resonances around the zero-mass frequencies. This feature led to enhanced micro-motion responses in non-overlapping frequency bands denoted by Δb_n as shown in **Figure R1b**, which enabled the MMR to behave as a mechanical frequency-division multiplexing system, achieving high sensitivity within a broad bandwidth. In essence, the core design of MMR is that configuring zero

effective mass in the unit cell results in a near-infinite effective piezoelectric coefficient, allowing access to micro-motion sensing with a high sensitivity around the zero-mass frequency. We further calculated the practical $d_{33\text{eff}}^n(\omega)$ through experimental verification. Results demonstrated that the practical $d_{33\text{eff}}^n(\omega)$ showed excellent agreement with the theoretical design and numerical simulation as shown in **Figure R1b**.

Figure R1 | Theoretical design, numerical simulation and experiments. a, Theoretical design of the frequency-dependent effective piezoelectric coefficients and zero effective masses. **b,** Numerical simulation and experimental verification of frequency-dependent effective piezoelectric coefficients.

We further verified the stability and reliability of MMR through experiments under different time. Considering that the sensor needs to be calibrated before working, we study the amplitude stability of the sensor during a continuous long-term measurement. Since MMR is composed of multiple unit cells, we take the first unit cell (zero-mass frequency of 436 Hz) as an example to analyze its stability and reliability. As shown in **Figure R2a**, we firstly apply a swept-frequency micro-motion excitation to the MMR from 0 Hz to 3500 Hz at the room temperature of 25°C, and the excitation time of a single cycle is 10 seconds. From the output results of the unit cell, it can be seen that the unit cell has a strong output at about 1.25 s because the frequency of the excitation signal at this time is near the zero-mass frequency of the MMR. And the average value of the whole output signal does not deviate from the zero-starting point, which means that the zero drift of the average amplitude is small. As shown in **Figure R2b**, to further quantify these zero drifts of average amplitude, we applied a cyclic swept-frequency excitation to the MMR over 10 cycles with a duration of 100 s. We then calculated the zero drift of the MMR under 10 cycles of excitation as shown in **Figure R2c**. From the calculation results, the MMR has a slight zero drift with a periodic amplitude less than 0.01 under the long-term cyclic excitation. This may be due to the low-frequency disturbance oscillation caused by the swept-frequency excitation, which can be eliminated and calibrated by a simple filtering algorithm in the computational multi-channel demodulation process.

Figure R2 | Stability and reliability under long-term measurement. a, Swept-frequency signal under single cycle. **b,** Swept-frequency signal under ten cycles. **c,** Zero drift of measured amplitude under ten cycles.

We also gave specific experimental data (with normalized amplitude) in the demonstrated remote-vibration monitoring, smart-driving assistance and structural health monitoring as shown in **Figure R3**. From the results as shown in **Figure R3a**, we gave the specific data of 7 Do-Ti scales and 7 higher Do'-Ti' scales in the demonstrated remote-vibration monitoring scenario. Each of the 14 scales lasts 0.5 seconds and is cycled twice. We then calculated the zero drift of the musical scale signal as shown in **Figure R3b**, from which we found that the magnitude of the zero drift is less than 0.01. Since the output amplitude is around 1, the ratio of the zero drift to the output amplitude is less than 1%, which demonstrated that MMR showed strong stability and reliability in the remote-vibration monitoring scenario. Moreover, we gave the specific experimental data of failed engine signal in the demonstrated smart-driving assistance. As shown in **Figure R3c** and **Figure R3d**, we calculated that the zero drift of the normal engine signal is less than 0.01. Similarly, the zero drift of the failed bearing signal in the demonstrated structural health monitoring is also less than 0.01 as shown in **Figure R3e** and **Figure R3f**. In summary, we gave specific experimental data in demonstrated remote-vibration monitoring, smart-driving assistance and structural health monitoring, which confirmed that MMR showed strong stability and reliability in the measurement of amplitude stability with a zero drift of less than 0.01.

With the above analysis, we demonstrated that experimental results showed excellent agreement with the theoretical design and numerical simulation. We verified that MMR shows strong stability and reliability in the demonstrated application scenarios including remote-vibration monitoring, smart-driving assistance and structural health monitoring. Overall, these experimental results demonstrate that the theoretical design and numerical simulation being applicable in practice is reliable.

Figure R3 | Specific data on stability and reliability in different application scenarios. a, Musical scale signal and its **(b)** amplitude stability. **c,** Failed engine signal and its **(d)** amplitude stability. **e,** Failed bearing signal and its **(f)** amplitude stability.

Second, for the improvement of measurement accuracy: as shown in **Figure R4**, we presented the measurement details of time-vary frequency components via one-time measurement and five repeated measurements. We constructed nine harmonic components of equal amplitude. The frequency values of nine harmonic components were around 437 Hz, 684 Hz, 964 Hz, 1250 Hz, 1540 Hz, 1900 Hz, 2130 Hz, 2480 Hz and 2880 Hz, respectively. We added background noise to the original multi-frequency signal to construct an experimental multi-frequency signal with an SNR of -20 dB. Each harmonic component appeared sequentially at 0.1 s intervals in the multi-frequency signal denoted by S_1 . Through computational sensing, we presented the one-time measurement result denoted by S_0 with MMR as shown in **Figure R4a**. We obtained the measurement error (S_0-S_1) from 0.450 s to 0.475 s as shown in **Figure R4b**. The average measurement error via one-time measurement was less than 3%, which was calculated by the amplitude ratio of the error to the original signal. Furthermore, we improve the accuracy of the testing data by averaging multiple repeated measurements. As shown in **Figure R4c**, we presented the measured result denoted by S_m with MMR via five repeated measurements. We obtained the measurement error (S_m-S_1) from 0.450 s to 0.475 s as shown in **Figure R4d**. The average measurement error via five repeated measurements was less than 1%. As a result, we demonstrate that the reconstruction accuracy of measured data can be further improved by repeated measurements. Compared with the one-time measurement, the measurement error of five repeated measurements with computational sensing is reduced from 3% to 1% in the demonstrated case.

Figure R4 | Improvement of measurement accuracy through repeated measurement. a, One-time measurement with **(b)** less than 3% error under -20 dB. **c**, Five repeated measurements with **(d)** less than 1% error under -20 dB.

Moreover, combining MMR with a deep learning technique, we demonstrated that the recognition accuracy of testing data can be further improved. As shown in **Figure R5**, the recognition accuracy is related to the signal-to-noise ratio (SNR) and the number of training samples. As shown in **Figure R5a**, the recognition accuracies with and without MMR both increased when the SNR is 0 dB, but the recognition accuracy without MMR was at most 70% and was always lower than that with MMR. This result proved that although the conventional sensor can still work at the SNR of 0 dB, its sensing performance was lower than MMR. When the SNR is -20 dB, the recognition accuracy without MMR was always below 40% even if the number of samples increased. We constructed four different signals according to the signal components shown in **Supplementary Table 5**, and completed verification experiments at -25 dB, -30 dB, -35 dB and -40 dB, respectively. The comparison of recognition accuracy under different SNRs with and without MMR is shown in **Supplementary Fig. 16**. On the whole, the recognition accuracy increases with the increase of the SNR, and increases with the increase of the number of training samples. As shown in **Figure R5b**, when the SNR is -25 dB and the number of training samples is 200, the recognition accuracy with MMR is 92.5%. Under the same conditions, we further increased the number of training samples from 200 to 400, and found that the recognition accuracy of testing data can be increased from 92.5% to 99.5% as shown in **Figure R5c**. Therefore, we demonstrated that the recognition accuracy of testing data can be improved by increasing the training samples in the constructed intelligent sensing system.

a

b

c

Figure R5 | Improvement of recognition accuracy by increasing training samples. a, Comparison of the recognition accuracy with and without MMR under different training samples. The recognition accuracy corresponding to **(b)** 200 and **(c)** 400 training samples under -25 dB.

In the revised supplementary information, we have added new results to **Supplementary Fig. 9** to better present the improvement of measurement accuracy through repeated measurement as follows:

Supplementary Fig. 9 | Comparison between the results measured with and without MMR. **a**, The clear signal components with MMR. **b**, The unclear signal components without MMR. **c**, The constructed original signal S_1 consisting of nine harmonic components. **d**, At a signal-to-noise ratio (SNR) of -20 dB, the measured signal S_2 without MMR has strong background noise. **e**, One-time measurement S_0 with MMR has **(b)** less than 3% error via computational multi-channel demodulation. **c**, Average of five repeated measurements S_m with MMR has **(d)** less than 1% error.

In the revised supplementary information, we have plotted a new **Supplementary Fig. 17** to better present the improvement of recognition accuracy by increasing the training samples as follows:

Supplementary Fig. 17 | Improvement of recognition accuracy by increasing training samples. The recognition accuracy corresponding to (a) 200 and (b) 400 training samples under -25 dB.

To clearly explain the improvement of measurement accuracy, we have added further analysis in **Supplementary Note 7** addressing this issue as follows:

“We presented the measurement details as shown in **Supplementary Fig. 9**. For the original multi-frequency signal, we constructed nine harmonic components of equal amplitude. The frequency values of nine harmonic components were around 437 Hz, 684 Hz, 964 Hz, 1250 Hz, 1540 Hz, 1900 Hz, 2130 Hz, 2480 Hz and 2880 Hz, respectively. We added background noise to the original multi-frequency signal to construct an experimental multi-frequency signal with an SNR of -20 dB. With the above workflow of signal reconstruction as shown in **Supplementary Fig. 8**, we obtained the time-frequency representation of the experimental multi-frequency signal with MMR. From the reconstructed signal spectrum with MMR as shown in **Supplementary Fig. 9a**, the measured result presented a clear time-frequency representation of multi-frequency components. In contrast, as shown in **Supplementary Fig. 9b**, the result with conventional piezoelectric mechanoreceptor presented unclear signal components and strong background noise, because the multi-frequency components with an SNR of -20 dB cannot reach the detection limit of conventional piezoelectric mechanoreceptor.

We further presented the time-domain details of the measured signal. As shown in **Supplementary Fig. 9c**, each harmonic component appeared sequentially at 0.1 s intervals in the multi-frequency signal denoted by S_1 . The measured result that was denoted by S_2 without MMR was shown in **Supplementary Fig. 9d**, from which we found that the measured signal contained strong background noise. Through computational sensing, we presented the measured result denoted by S_0 with MMR in **Supplementary Fig. 9e**. We obtained the measurement error ($S_0 - S_1$) from 0.450 s to 0.475 s as shown in **Supplementary Fig. 9f**. The average measurement error was less than 3%, which was calculated by the amplitude ratio of the error to the original signal. Furthermore, we improve the accuracy of the testing data by averaging multiple repeated measurements. As shown in **Supplementary Fig. 9g**, we presented the measured result denoted by S_m with MMR via five repeated measurements. We obtained the measurement error ($S_m - S_1$) from 0.450 s to 0.475 s as shown in **Supplementary Fig. 9h**. The average measurement error via five repeated measurements

was less than 1%. As a result, we demonstrate that the reconstruction accuracy of measured data can be further improved by repeated measurements. Compared with the one-time measurement, the measurement error of five repeated measurements with computational sensing is reduced from 3% to 1% in this demonstrated case. From the measured results and error analysis, we found that the measured results with MMR were relatively close to the original signal even in an SNR of -20 dB. Thus, we achieved highly sensitive and broadband micro-motion sensing in MMR that is inaccessible in the conventional piezoelectric mechanoreceptor.” (Supplementary Note 7)

To clearly explain the improvement of recognition accuracy, we have added further analysis in **Supplementary Note 13** addressing this issue as follows:

“Considering that the mechanical micro-motions can be different with different styles of vibration and energy levels, we evaluated the lowest signal-to-noise ratio (SNR) at which MMR was able to detect micro-motion signals. As shown in **Supplementary Note 11**, we constructed an intelligent sensing system by combining MMR with a deep learning technique. The micro-motion signal was measured by MMR and then was transported to a deep neural network to complete the signal recognition. Here, we validated the performance of MMR by comparing the recognition rates under different SNRs. From **Fig. 4b** of the main text, when the SNR was -20 dB, we have demonstrated that the recognition accuracies of patterns with and without MMR were about 99.5% and 34.0%, respectively. We constructed four different signals according to the signal components shown in **Supplementary Table 5**, and completed verification experiments at -25 dB, -30 dB, -35 dB and -40 dB, respectively. The comparison of recognition accuracy under different SNRs with and without MMR is shown in **Supplementary Fig. 16**. From the results, we found that the recognition accuracy with MMR decreases as the SNR decreases. Specifically, when the SNRs are -20 dB and -40 dB, the recognition accuracies with MMR are 99.5 % and 58.0%, respectively. This is because the signal components that can be measured by MMR decrease as the SNR decreases. From the results without MMR, we found that the recognition accuracy without MMR fluctuates below 40% as the SNR decreases. This is because the conventional piezoelectric mechanoreceptor cannot measure clear signal components since the signal with SNR below -20 dB cannot reach the detection limit.

We further demonstrated the improvement of recognition accuracy by increasing training samples. As shown in **Supplementary Fig. 17a**, when the SNR is -25 dB and the number of training samples is 200, the recognition accuracy with MMR is 92.5 %. Under the same conditions, we further increased the number of training samples from 200 to 400, and found that the recognition accuracy of test data can be increased from 92.5% to 99.5% as shown in **Supplementary Fig. 17b**. Therefore, we demonstrated that the recognition accuracy of testing data can be improved by increasing the training samples in the constructed intelligent sensing system.” (Supplementary Note 13)

In addition, the major points discussed above have been included in the revised manuscript addressing this issue as follows:

“First, we constructed harmonic signals with nine frequency components of equal amplitude as the multi-frequency signal. From the measured results (Fig. 3c), the maximum sensitivity of MMR was improved by two orders of magnitude from $0.812 \text{ mv m}^{-1}\text{s}^2$ to $36.540 \text{ mv m}^{-1}\text{s}^2$ as compared to the conventional piezoelectric mechanoreceptor. The measurement accuracy can be further improved through multiple repeated measures (Supplementary Note 7 and Supplementary Fig. 9). The lowest detection limit of MMR was demonstrated to reach 10^{-6} g in the order of magnitude (Supplementary Note 8).” (Page 9, Line 207)

“By matching the integrated central processing unit (CPU), MMR can be packaged as an all-in-one system with sensing, storage and computing functions (Supplementary Note 19), which can enable the entire sensing system to be simpler and more portable⁶⁸⁻⁷⁰. MMR can also provide high versatility for complex micro-motion scenes with higher accuracy by incorporating advanced artificial intelligence (AI) techniques⁷¹.” (Page 14, Line 312)

We would like to thank the reviewer for the careful and constructive review on our work again. In the revised manuscript and supplementary information, we have carefully considered the reviewers’ suggestions and comments and provided corresponding revisions, hoping that these revisions are satisfactory.